# Regulation of dynamic pigment cell states at single-cell resolution

Margherita Perillo[1], Nathalie Oulhen[1], Stephany Foster[1], Maxwell Spurrell[1], Cristina Calestani[2], Gary Wessel[1]*

[1]Department of Molecular and Cellular Biology Division of Biology and Medicine Brown University, Providence, United States; [2]Department of Biology Valdosta State University, Valdosta, United States

**Abstract** Cells bearing pigment have diverse roles and are often under strict evolutionary selection. Here, we explore the regulation of pigmented cells in the purple sea urchin *Strongylocentrotus purpuratus,* an emerging model for diverse pigment function. We took advantage of single cell RNA-seq (scRNAseq) technology and discovered that pigment cells in the embryo segregated into two distinct populations, a mitotic cluster and a post-mitotic cluster. *Gcm* is essential for expression of several genes important for pigment function, but is only transiently expressed in these cells. We discovered unique genes expressed by pigment cells and test their expression with double fluorescence in situ hybridization. These genes include new members of the *fmo* family that are expressed selectively in pigment cells of the embryonic and in the coelomic cells of the adult - both cell-types having immune functions. Overall, this study identifies nodes of molecular intersection ripe for change by selective evolutionary pressures.

*For correspondence:
rhet@brown.edu

Competing interests: The authors declare that no competing interests exist.

## Introduction

Biologically crafted pigments are nearly ubiquitous in nature. Pigmentation can reveal identity, protection, and even have antimicrobial functions. Pigmentation in an individual or in a species can change annually, or even instantaneously for camouflage for exmple arctic hares or cephalopod mollusks and chameleons respectively, making pigmentation even more dynamic in response to the environment. Echinoderms, a phylum of marine organisms, display diverse pigmentation schemes in embryos, larvae, and adults. From an intense red sea star (*Fromia milleporella*) to a black and white striped brittle star (*Ophiactis savignyi*) to the variegated sea urchin (*Lytechinus variegatus),* pigment is a pervasive feature of this phylum. Because of the molecular techniques now available for echinoderms, mechanisms controlling pigmentation in these animals are being revealed (*Hira et al., 2020*; *Liu et al., 2019*; *Wessel et al., 2020*; *Yaguchi et al., 2020*). The purple sea urchin, *Strongylocentrotus purpuratus*, provides a molecularly tractable model organism to dissect the developmental importance of pigmented cells, and the biosynthesis of their pigment. The adult of this echinoderm is intensely dark purple as a result of a family of polyketides, including echinochrome and various related spinochromes (*Anderson et al., 1969*). Furthermore, like most sea urchins, *S. purpuratus* larvae are pigmented due to the accumulation of a red/orange pigment in single cells embedded in, and scattered throughout, the aboral ectodermal layer (*Gibson and Burke, 1985*; *Griffiths, 1965*; *Gustafson and Wolpert, 1967*; *Kominami et al., 2001*; *McClendon, 1912*). This pigment is a napthoquinone called echinochrome A, which accumulates in the pigment cell precursors during gastrulation in *S. purpuratus* (*Calestani et al., 2003*; *Griffiths, 1965*; *Kuhn and Wallenfels, 1940*; *Oulhen and Wessel, 2016*). Mutations that affect the pigmentation pathway lead to albinism (*Calestani et al., 2003*; *Oulhen and Wessel, 2016*; *Wessel et al., 2020*), and adult sea urchins that lack pigments are less resistant to environmental challenges (*Wessel et al., 2020*). The functional

relationship between these larval and adult pigments and associated cells, and whether their biosynthetic pathways are similar, are open questions.

A known function of sea urchin larval pigment cells includes an essential role in the innate immune defense system (*Buckley and Rast, 2017*; *Ch Ho et al., 2016*; *Hibino et al., 2006*; *Hira et al., 2020*; *Kiselev et al., 2013*; *Ransick and Davidson, 2006*; *Schrankel et al., 2016*; *Solek et al., 2013*). When larvae are exposed to bacteria, pigment cells migrate from the ectoderm to the gut, a site for invading microbes, and interact with other immune cells (*Ch Ho et al., 2016*). This cell-cell interaction is at least in part regulated by IL17 cytokine (*Buckley and Rast, 2017*). Echinochrome A is also present in eggs of certain sea urchin species, in immune cells of the coelomic fluid of the adult (the red spherule cells, RSC), in spines, gonads, the digestive system, and in tube feet (*Brasseur et al., 2018*; *Coates et al., 2018*; *Johnson, 1969*; *Perry and Epel, 1981*; *Service and Wardlaw, 1984*; *Smith et al., 2018*; *Smith et al., 2010*). It is also thought that pigment is released by the pigmented cell that directly harms microbes. The antimicrobial mechanism of echinochrome has not been completely resolved, but evidence suggests its production of hydrogen peroxide and/ or iron chelation, abates microbial proliferation, (*Coates et al., 2018*; *Lebedev et al., 2005*; *Perry and Epel, 1981*). All of these studies agree that sea urchin pigments have anti-microbial activity, and that these small molecules may also contribute to states of cell physiology and gene expression (*Jeong et al., 2014*; *Kim et al., 2018*).

The developmental origins of pigment cells in the purple sea urchin have been traced to a group of mesodermal cells, the non-skeletogenic mesoderm (NSM) (*Cameron et al., 1991*; *Croce and McClay, 2010*; *Materna and Davidson, 2012*; *McClay et al., 2000*; *Oliveri et al., 2002*; *Ruffins and Ettensohn, 1996*; *Sherwood and McClay, 1999*; *Sweet et al., 1999*). Among the NSM cell types, pigment cells are specified first by Delta/Notch (D/N) signaling from the micromeres (*Calestani et al., 2003*; *Calestani and Rogers, 2010*; *Croce and McClay, 2010*; *Davidson et al., 2002a*; *Foster et al., 2020*; *Materna and Davidson, 2012*; *McClay et al., 2000*; *Oliveri et al., 2002*; *Ransick et al., 2002*; *Rast et al., 2002*; *Sherwood and McClay, 1999*; *Sweet et al., 2002*; *Sweet et al., 1999*). The D/N signaling directly activates the transcription factor *glial cells missing* (*gcm*), which in turn activates several pigment cell genes, including: *polyketide synthase1* (*pks1*), three *flavin monooxygenases* (*fmo1, 2 and 3*), a *sulfotransferase* (*sult*) and a *dopachrome tautomerase* (*dopt*) (*Calestani et al., 2003*; *Calestani and Rogers, 2010*; *Davidson et al., 2002b*; *Ransick and Davidson, 2006*; *Ransick et al., 2002*; *Rast et al., 2002*). The enzymes Pks1 and Fmo1 (reannotated as Fmo3 in this study) are required for the biosynthesis of the echinochrome in pigment cells (*Calestani et al., 2003*). Pks1 is required for any pigment biosynthesis in the adult while Fmo3 is required for accumulation of Spinochrome B in spines, but not for echinochrome synthesis in RSC and tube feet (*Wessel et al., 2020*). Genes expressed selectively in pigment cells, such as *gcm* and *pks1,* are first detected at blastula stage in a ring of NSM precursors surrounding the skeletogenic mesenchyme (SM) and later during blastula stage they become restricted to the aboral NSM (*Calestani et al., 2003*; *Duboc et al., 2010*; *Gibson and Burke, 1987*; *Gustafson and Wolpert, 1967*; *Kominami et al., 2001*; *Materna et al., 2013*; *Ransick and Davidson, 2012*; *Ransick et al., 2002*). At the gastrula stage, *gcm,* as well as the pigment biosynthetic genes are expressed in cells migrating into the blastocoel, and their expression is maintained throughout larval development (*Calestani et al., 2003*; *Gibson and Burke, 1987*; *Gustafson and Wolpert, 1967*; *Kominami et al., 2001*; *Ransick and Davidson, 2006*; *Ransick et al., 2002*). Some evidence suggests *gcm* may be required for the development of pigment cells, since in *gcm* knockdown embryos, the cells do not migrate to the ectoderm at the mesenchyme blastula stage, and pigments are lost (*Ransick and Davidson, 2006*). Moreover, ectopic expression of *gcm* in skeletogenic mesoderm (SM) cells is sufficient to induce a similar single cell migration to the ectoderm and the accumulation of pigment in these cells, but not the acquisition of the typical morphology of pigment cells (*Damle and Davidson, 2012*). *Gcm* is also expressed at higher levels in adult hematopoietic tissues, the pharynx and the axial organ, which produce new coelomocytes upon immune challenge (*Golconda et al., 2019*). Thus, while *gcm* is an important transcriptional regulator, its mechanistic role in larval and adult pigmented cells is still unclear.

Here we tested the role of *gcm* in pigment cell development through single cell transcriptomic approaches, and define polytypic pigment cell subpopulations. Our analyses revealed two distinct groups of pigment cells, a mitotic population and a differentiated population. We further analyzed the diversity of *fmo* genes, essential players in the pigmentation pathway, and found a group of *fmo*

genes specifically expressed in the embryonic pigment cells and in the coelomocytes of the adult immune system. We also performed a single cell mRNA analysis of *gcm* morphants and revealed that pigment cell specification is dependent on *gcm*. Our study leverages our understanding of the Gcm transcription factor, an essential factor in the development of pigmentation in the larvae (*Calestani and Rogers, 2010*; *Oulhen and Wessel, 2016*; *Ransick and Davidson, 2006*; *Ransick and Davidson, 2012*; *Wessel et al., 2020*) and adult (*Wessel et al., 2020*) and provides deep datasets for exploring evolutionary selection of the biosynthesis of pigmentation.

## Results

### The ectodermal transcriptional state at a single cell level

Pigment cells of the sea urchin larva appear as a homogeneous population based on their morphology, expression of the transcription factor *gcm*, the enzyme *pks1*, and the presence of red pigment. Yet the many immune functions ascribed to pigmented cells in larvae and in adult tissues implies a heterogeneous population of cells. We tested this hypothesis with single cell sequencing technology. To enrich for pigment cells, we dissociated embryos/larvae from 48hpf and 72hpf, enriching for the ectodermal layer that includes the pigment cells (*Calestani et al., 2003*; *McClay and Marchase, 1979*; *Ransick and Davidson, 2012*), captured cells by drop-seq technology (*Figure 1A*), and sequenced the resulting cDNAs at a single cell level. The ectoderm-enriched single cell transcriptome revealed multiple cell-types, mostly of ectodermal origin as expected, and the overall cell cluster organization was highly reproducible between 48hpf and 72hpf samples (*Figure 1B*). The cells that formed the main clusters of the tSNE plots represented ectodermal cell types, including ciliary band cells, apical, oral, aboral, and lateral ectodermal cells (*Figure 1C*) based on expression of known cell-type marker genes. The pigment cell markers *gcm* and *pks1* were found in clusters 2 and 13, which showed a similar transcriptome profile, and cluster 12, which had a transcriptomic profile that was distinct from clusters 2 and 13. In addition to *gcm* and *pks1*, cluster 12 also expressed the transcription factor *six1* and its co-factor *eya*, which are expressed in aboral secondary mesoderm at mesenchyme blastula (*Poustka et al., 2007*; *Ransick and Davidson, 2012*). Cluster 2 and 13 expressed *gcm*, *pks1* and well-known pigment cell markers (*fmo3, fmo5-1, sult1c2* [*Calestani et al., 2003*; *Ransick and Davidson, 2006*; *Ransick and Davidson, 2012*; *Ransick et al., 2002*]). Other key clusters we found are the proneural apical plate (cluster 10), ciliary band neurons (cluster 16), serotonergic neurons (cluster 15 and 17), skeletal cells (cluster 11), and mid-gut cells (cluster 14). Overall our single cell transcriptomes showed consistent cell types between 48hpf gastrulae and 72hpf early larvae.

### *Gcm* is enriched in three clusters at Gastrula and early larva stages

Single cell transcriptomes revealed cryptic cell-state distinctions by assessing their individual gene expression profiles, even though the limiting mRNA captured in each drop likely over-represents the more abundant transcripts in a cell. With sufficient depth in sequencing, these datasets allowed us to identify cell-state specific marker genes and to predict the nature of each cluster (details in *Figure 1* legend). We use the term cell *state* here instead of cell *type* for the compelling reason that distinctions in these cell groupings by this technology does not always mean a distinction in cell morphology, physiology, function, or cellular location, key characteristics of the use of cell *type* (*Clevers et al., 2017*). Transcript profiles outnumber the classic definition of cell types, which is necessarily limited when one considers lineage variations and boundaries between lineages. Instead, cell *states* refer to populations of cells that show distinctions in transcript accumulation, which may or may not be reflected in the end–point function of the cell.

First, we analyzed the cell state of *gcm* expression during embryonic development, from the early zygote to the larva stage. To this aim we integrated the three *gcm*-enriched clusters from the ectodermal dataset (2, 12 and 13) to the *gcm*-enriched cluster of a time-course single cell analysis of sea urchin embryos encompassing eight developmental time points, from eight-cell stage to late gastrulae (*Foster et al., 2020*). The *gcm* enriched cluster from this dataset was integrated to that of the ectodermal datasets and analyzed together as in Seurat V3 (*Stuart et al., 1821*). This analysis revealed 7 cell states of *gcm*-expressing cells, across the developmental time points (*Figure 2A*, *Figure 2—figure supplement 1*), showing that *gcm* is dynamically expressed between many different

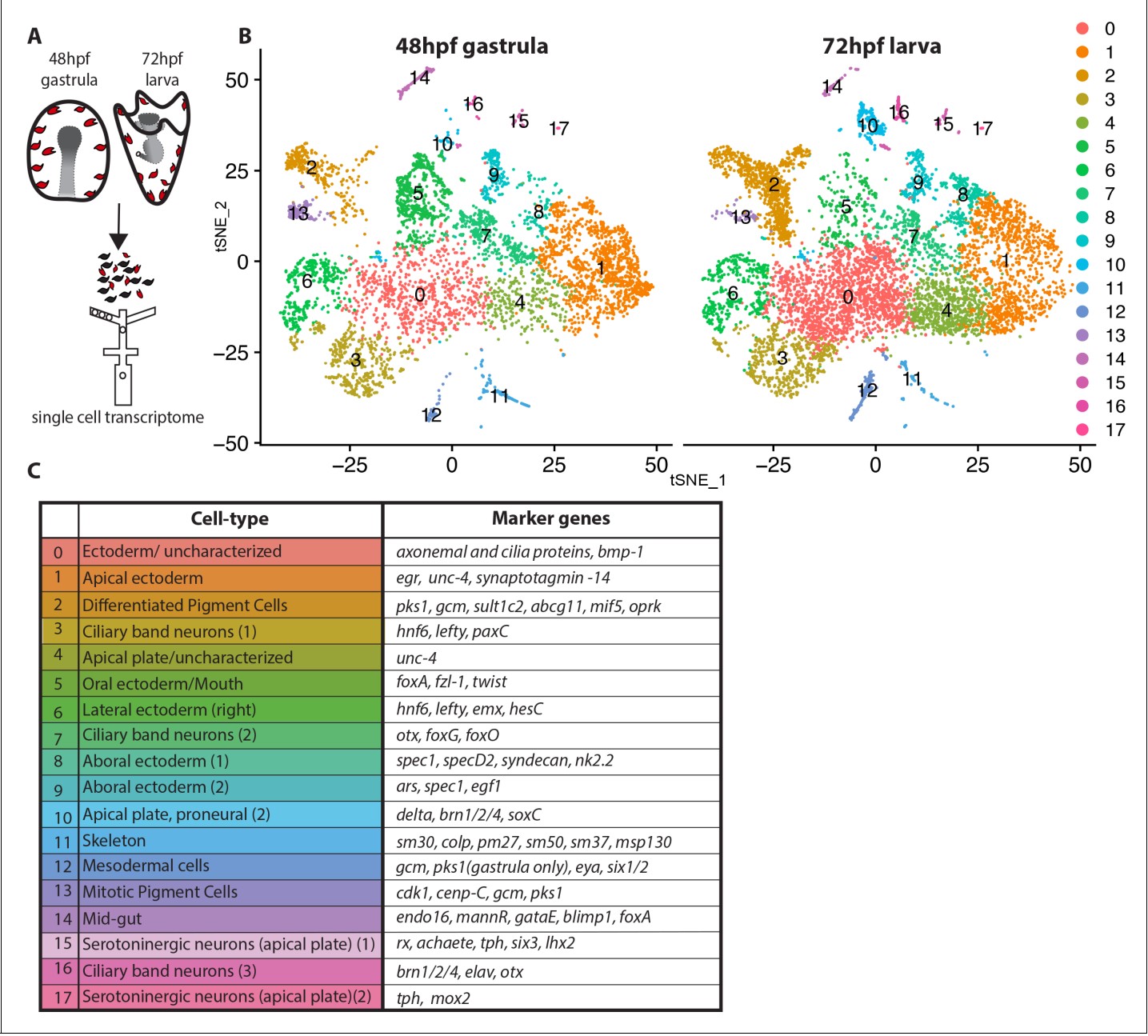

**Figure 1.** Dissection of *S. purpuratus* ectodermal cell-types by scRNA-seq. (**A**) Cartoon summarizing the dissociation of gastrulae and larvae (pigment cells in red, other cell-types in black), cell separation through drop-seq technology and RNA sequencing. Our single cell sequence datasets detected 20,489 reads/genes and a median 2,909 UMI counts per cell at 48hpf as well as 19,716 reads/genes and a median 1,136 UMI counts per cell at 72hpf. (**B**) tSNE plots of gastrula (48hpf) and larvae (72hpf) enriched for the ectodermal cell-types. Colors indicate major cell-types grouped by gene expression similarity. In the 48hpf sample, 5688 single cells were captured for downstream analysis, sequenced at a level of 81,121 reads per cell. The 72hpf sample includes 8178 single cells with an average of 54,788 reads per cell. The two samples were integrated to identify conserved cell types and cluster marker genes using Seurat. (**C**) Table summarizing the cell-type for each cluster (48hpf +72hpf) with the most representative marker genes (colors in the table reflect colors of the tSNE plot clusters).

cell states through development. This is an unusual feature for gene expression in that *gcm* is not just accumulating in more cell states as they appeared, but that gcm is *transiently* expressed in many different cell states. This suggests that *gcm* in this embryo may initiate various gene activities, but that maintenance of that gene activity is dependent on other transcriptional mechanisms. This may reflect the many different activator and repressor roles that *gcm* has in this embryo.

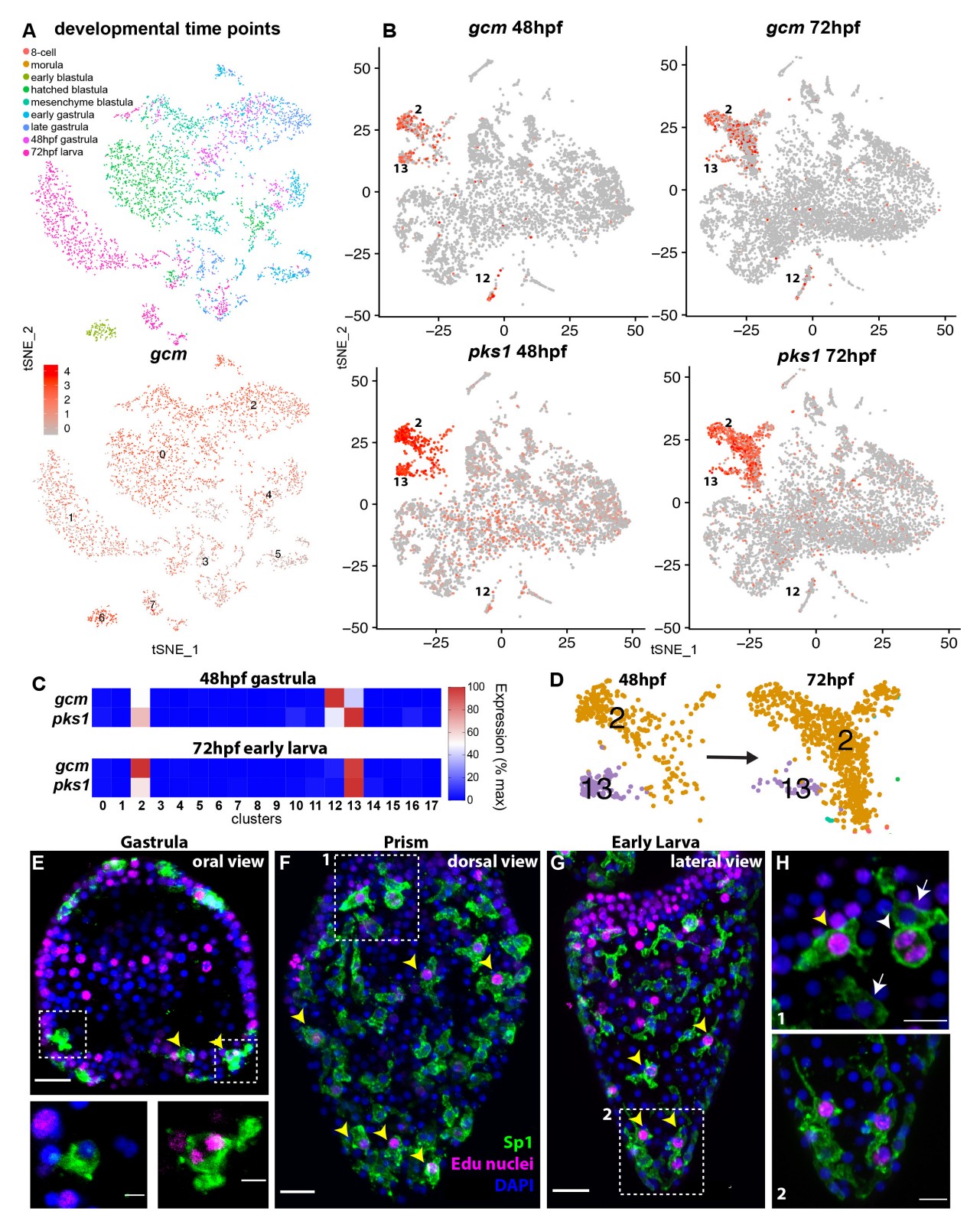

**Figure 2.** *gcm* marks pigment cell mitotic and post-mitotic populations. (**A**) tSNE plot showing integration of gcm enriched clusters from datasets encompassing nine time points across sea urchin development: 8 cell stage, morula, early blastula, hatched blastula, mesenchyme blastula, early gastrula, late gastrula, 48hpf, and 72hpf. Cells are colored by time point. Feature plots for selected genes are provided in *Figure 2—figure supplement 1*. Using the dataset of *Foster et al., 2020* we found that Gcm was first detected at 64 cell stage, when there are nine *gcm*+ cells, 1% of

*Figure 2 continued on next page*

*Figure 2 continued*

total cells. At morula stage 125 cells express *gcm*, 3.48%. Early blastula:13.9% Hatched: 9.5% Mesenchyme blastula: 7.6% Early gastrula: 6.5% and late gastrula 4.7%. (B) Feature plots of gastrulae and larvae colored for gcm or pks1 showing gene expression in clusters 2 and 13 at 48hpf and 72hpf, and cluster 12 at 48hpf. Violin plots showing expression of *gcm* and *pks1* are in *Figure 2—figure supplement 2*. (C) Heatmap from scRNA-seq data represents expression of *gcm* and *pks1* in the three clusters mitotic (13) and differentiated (2) pigment cells, and mesodermal cells (12). *gcm* expression is high in the mesodermal cluster at 48hpf, but decreases at 72hpf. (D) Magnification of tSNE plot from *Figure 1B* shows that the number of cells in the differentiated pigment cells cluster (2) increases over time, while the mitotic cluster (13) decreases. (E,F,G) Double staining for pigment cells (Sp1 antibody) and Edu labeling to mark cells that have recently synthesized DNA (yellow arrowheads). Note that mitotic pigment cells are broadly distributed within the embryos and larvae. (H) Magnifications of boxes 1 and 2 from figure F and G. White arrowheads show a pigment cell undergoing mitosis, yellow arrowhead shows another mitotic pigment cell, arrows show pigment cells that are non-mitotic. Scale bars are 100 µm (F, G), 20 µm (E) and 5 µm (H). All images are stacks of merged confocal Z sections.

The online version of this article includes the following figure supplement(s) for figure 2:

**Figure supplement 1.** Examples of cell states across developmental stages.

**Figure supplement 2.** Violin plots showing expression of pigment cell marker genes across different clusters at gastrula and early larval stage.

We then analyzed the expression of *gcm* in two time points key in pigment cell differentiation (48 and 72hpf). Of these cell clusters expressing *gcm*, we found enrichment for other pigment cell terminal differentiation genes in three clusters in late gastrulae (48hpf; clusters 2,12, 13) and two clusters in larvae (72hpf; clusters 2, 13 shown in *Figure 2B,C*). We next examined pigment gene expression in these three clusters. *Pks1* was highly expressed in cells of clusters 2 and 13, and at low levels in cells of clusters 12 at gastrula stage (see violin plot, *Figure 2—figure supplement 2*). Feature plots showed that clusters 2 and 13 are spatially close, but separate, meaning that cell states 2 and 13 had similarities in their transcript profiles, while cluster 12 diverged significantly from cluster 2 and 13 (*Figure 2B*). To define the differences in each of these three *gcm+* clusters we performed differential gene expression analysis (clusters 2 vs 12, 2 vs 13 and 12 vs 13) and analyzed the average logFC for *pks1* in each cluster. We found that *pks1* was highly expressed in clusters 2 and 13 with respect to cluster 12, where *pks1* levels were low. To determine if the *pks1* low levels are stage dependent we normalized the data and compared cluster 12 to cluster 13 at 48hpf and 72hpf separately and found that at both stages *pks1* levels in cluster 12 were lower than in clusters 2 and 13 (*Supplementary file 1*). Similarly, other known pigment cell markers, such as *sult1c2* or *fmo5-1* (previously called *sult* and *fmo2*; *Calestani et al., 2003*; *Ransick and Davidson, 2012*) were highly expressed in clusters 2 and 13 but not in cluster 12 (*Supplementary file 1*). This finding shows that *gcm* is dynamically expressed during embryogenesis, and that it transiently marks two distinct pigment cell clusters (2 and 13). Another cluster that included mesodermal markers (12) expressed *gcm* in gastrulae, and low *pks1* expression, suggesting that this is a population of pigment cell mesodermal precursors.

### *Gcm* marks mitotic and post mitotic pigment cell populations

Having defined that clusters 2 and 13 are related based on their transcriptomic profile, and represent pigment cells, we analyzed these clusters in additional detail. The number of cells in these two clusters changed from 48hpf to 72hpf. Over a period of 24 hours, the proportion of cells in cluster two increased, while the proportion of cells in cluster 13 decreased (*Figure 2D*). To investigate the differences between the clusters, we performed differential gene expression analysis of clusters 2 and 13 at 48hpf and 72hpf. Both clusters expressed pigment cell markers at high levels, and the only difference was that cluster 13 (the cluster with fewer cells at both time points) also expressed transcripts encoding S-phase and mitotic activities at high levels. The elevated markers in cluster 13 included *cdk1*, *pcna*, DNA polymerases, DNA ligases, condensins, and centromere proteins (*Supplementary file 2*), all genes that were absent in cluster 2. This result suggested that there were two pigmented cell clusters with a different cell cycle potential, cluster 13 being pigment cells still undergoing mitosis and cluster two being post-mitotic pigment cells. To test whether there are pigment cells in a mitotic state, or whether migrating pigment cells are all mitotic, we performed an EdU pulse for 30 min, followed by Sp1 staining (a conserved pigment cell marker [*Gibson and Burke, 1985*]). We found that from gastrula to early larva, cells that were in S-phase represented 34% of total pigment cells, and were broadly distributed in the embryos (*Figure 2E,F,G*). These cells were synthesizing DNA and dividing (see anaphase cells, inset *Figure 2E*, and *Figure 2H*, white

arrowhead). This finding of a mitotic pigment cell cluster suggests that pigment cells have the ability to divide as migratory cells. Unique about this tSNE cluster of pigment cells though is the significant increase in mRNAs involved in cell cycle progression, a feature not normally seen in other dividing cells within the embryo.

## Expression of unique pigment genes identified by scRNA-seq

To test for polytypic pigment cells, we defined the expression of newly identified pigment cell markers found in the scRNA-seq by in situ hybridization. scRNA-seq data showed that *gcm* and *pks1* are expressed in the same group of cells at these timepoints. The co-expression of these two genes was always assumed based on a similar pattern of expression and functional studies (*Calestani et al., 2003*; *Calestani and Rogers, 2010*; *Materna et al., 2013*; *Ransick and Davidson, 2006*; *Ransick and Davidson, 2012*), but never tested by RNA co-localization. We independently tested the scRNA-seq result by performing double fluorescent in situ hybridization (FISH) of *gcm* and *pks1*. In blastulae and gastrulae, *pks1* and *gcm* were expressed in the same cells that have pigment cell morphology (*Figure 3A–B''*). In larvae, however, *pks1* expression was maintained in pigment cells, while *gcm* expression transitioned into the left coelomic pouch, the site of the future adult rudiment (*Figure 3C–C''*). Previously, one single *gcm*-expressing cell was found close to the coelomic pouch with a *gcm::gfp* recombinant BAC (*Ransick and Davidson, 2012*). We therefore used *gcm* in gastrulae and *pks1* in larvae as markers to assess whether the putative pigment cell genes were expressed exclusively in this cell population.

We then selected a few genes whose expression were relevant to pigment cell function and analyzed their expression pattern together with *gcm* (at gastrula, 48hpf) and *pks1* (at early larva, 72hpf) by double in situ hybridization. Among these genes, ABCG11 is an ATP-binding cassette transporter homolog of the *white* half-transporter that transports pigment precursors and intermediates into the *Drosophila* eye (*Mackenzie et al., 2000*; *Shipp and Hamdoun, 2012*). *Abcg*11 was selectively expressed in the pigment cell clusters 2 and 13, but not in cluster 12 (violin plots in *Figure 2—figure supplement 2*). Double FISH showed that Abcg11 was exclusively expressed in *gcm* and *pks1* + cells (*Figure 3D,E–E', F–F'*).

MIF (macrophage inhibitory factor) is a member of a gene family of cytokines that regulate innate immunity (*Hibino et al., 2006*; *Nishihira, 2000*). Of these genes, *mif5* was expressed in the same *gcm*+ pigment cell clusters at 48hpf (2, 12, and 13, violin plots in *Figure 2—figure supplement 2*), and by 72hpf it became enriched in the pigment cell clusters 2 and 13 exclusively. Double FISH confirmed expression of *mif5* in pigment cells at both stages (*Figure 3G–H'*). A known pigment cell marker is the enzyme sulfotransferase (*Calestani et al., 2003*). Single cell data showed that *sult1c2* was highly expressed in pigment cell clusters at 48hpf and 72hpf and violin plots showed that *sult1c2* expression remained low in a few cells of all the other cell types (violin plots in *Figure 2—figure supplement 2*). Double FISH confirmed a strong expression of *sult1c2* in the *gcm/pks1*+ cells (*Figure 3I–I'*). Opioid receptor, kappa 1-like (*oprk*) is a Cholecystokinin Receptor Type A, a G-protein coupled receptor that binds to cholecystokinin peptide hormones to modulate feeding and dopamine-induced behavior (*Crawley, 1991*). scRNA-seq data showed that *oprk* was enriched in clusters 2, 12 and 13 at 48hpf, but by 72hpf it was enriched also in the ciliary band neurons (cluster 16), and we detected a spotty expression in this region by FISH (arrowhead in *Figure 3L'* and violin plots in *Figure 2—figure supplement 2*). Lastly, we analyzed the expression profile of the gene glutamate receptor 6 (*gluR6*). *GluR6* was not detectable by single cell sequencing at 48hpf, and double FISH analysis showed only faint signals in *gcm*+ cells (violin plots in *Figure 2—figure supplement 2*; *Figure 3M–M'*). *GluR6* was detectable at 72hpf though at low levels in cluster 2 (differentiated pigment cells) and cluster 16, a cluster of neurons. Double FISH with *pks1* at 72hpf showed that *gluR6* was expressed in *pks1*+ cells in or underneath the ectoderm (*Figure 3M–M'* and inset one in N') and in cells that are not pigment cells also located in the blastocoel beneath (basal to) the ectoderm (*Figure 3N'* insets 1 and 2). *GluR6* was also highly expressed in both coelomic pouches (*Figure 3N–N'* and inset), a tissue not included in the scRNA-seq enriched for the ectoderm. Altogether the single cell transcriptome data and double FISH on the transcripts found in the scRNA-seq dataset indicated that genes for pigment cell function (besides *gcm* and *pks1* in gastrulae) are never expressed in the mesodermal cluster (cluster 12), but rather in the mitotic (cluster 13) and post-mitotic (cluster 2) pigment cell clusters.

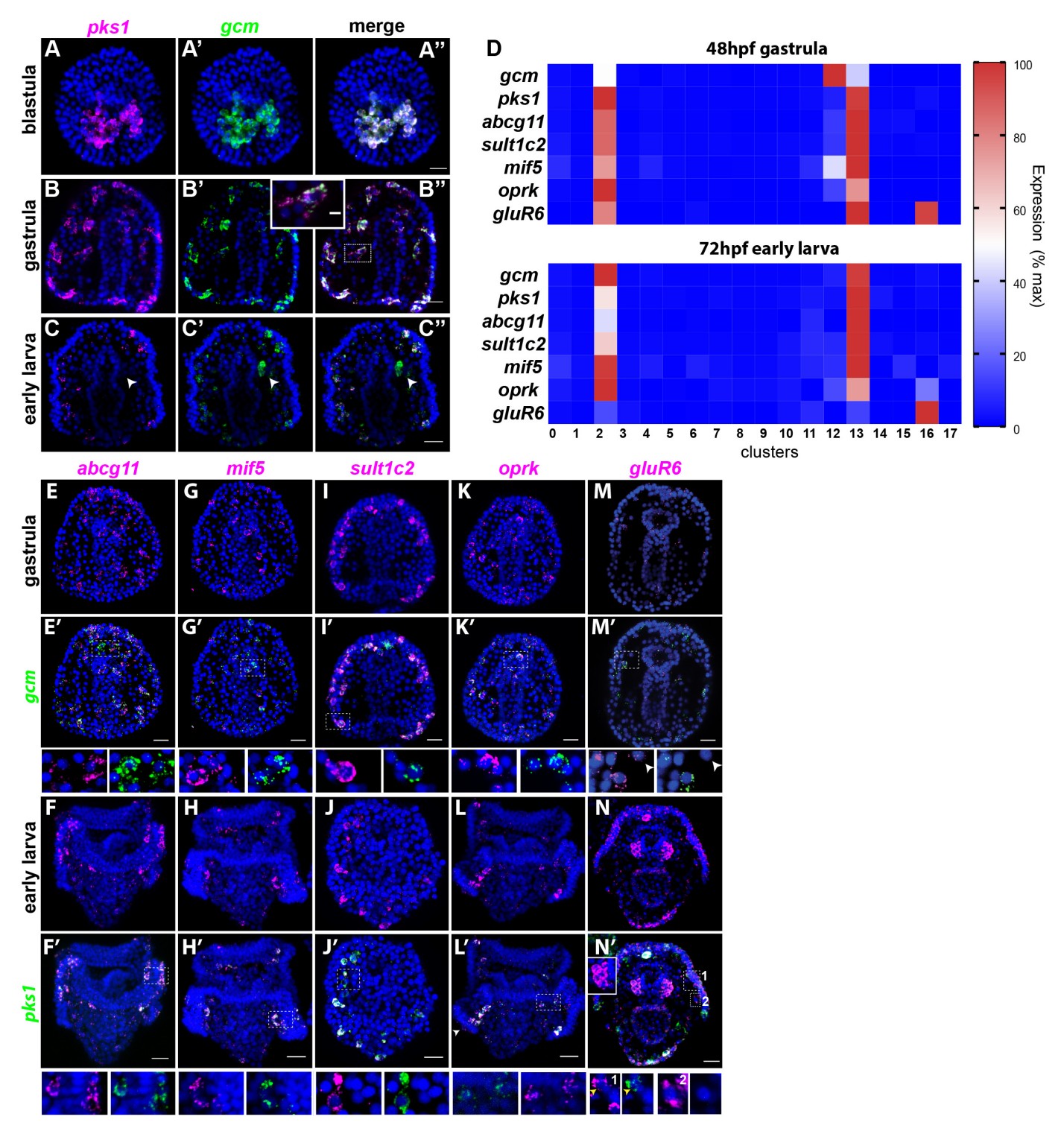

**Figure 3.** Gene expression in the pigment cells revealed by single cell RNAseq. (**A–C''**) Double fluorescent in situ hybridization (FISH) of the pigment cell markers *gcm* and *pks1*. The *pks1* and *gcm* genes are expressed in the same pigment cells in blastulae and gastrulae. Dashed line box indicates the cell magnified in B'' showing that mRNA for the two genes is expressed in the cell. In larvae, *gcm* is expressed in pigment cells and in the left coelomic pouch (white arrowheads). Nuclei are stained in blue (DAPI). (**D**) Heat map of pigment genes. At 48hpf, *gcm* is highly expressed in cluster 12, and the other genes are expressed at intermediate levels. At 72hpf *gcm* expression in cluster 12 decreased and also the expression of the other pigment genes decreased. *GluR6* and *oprk* are also expressed in cluster 16, ciliary band neurons. (**E–N'**) Double FISH of identified pigment genes with either *gcm* or *pks1* to mark pigment cells in gastrulae and larvae, respectively. Dashed line boxes indicate cells magnified below each picture. Violin

*Figure 3 continued on next page*

*Figure 3 continued*

and feature plots for the pigment cell genes are shown in *Figure 2—figure supplement 2*. (M–M'). *GluR6* expression in gastrulae is low. In larvae, *gluR6* is expressed in pigment cells (yellow arrowheads in J' inset 1) and in neurons (cluster 16) that appear to be lateral ganglions (inset 1 and 2 white arrowheads in N'). *GluR6* is also expressed in both coelomic pouches (inset in N'). Nuclei are depicted in blue (DAPI). Scale bars 50 µm, inset in B'-B'' scale bar 20 µm. All images are stacks of merged confocal Z sections.

## A diverse family of *fmos* is present in distinct pigment cell states

Having defined that there are two distinct sub-populations of pigment cells in embryos that expressed gcm and *pks1*, we investigated the expression of a class of enzymes that at least one representative modifies the polyketide pigment scaffold, the Flavin-dependent monooxygenases (Fmo). Fmos are present throughout phylogeny, and although their conserved protein domain is typical of proteins with oxidizing activity, it is still unclear the exact role in the sea urchin pigment biosynthetic pathway/s. To determine the expression of *fmos* in echinoderms, we sought all Fmo domains in the *S. purpuratus* genome (echinobase.org, V4.2; RRID:SCR_013732), and we found that not all of the identified *fmos* were specifically expressed in pigmented cells. We performed a phylogenetic analysis to look for protein subfamilies, and found 4 Fmo clades: one is in the same clade with vertebrate Fmo proteins (clade 2), two are not closely related to Fmo proteins in other organisms (clades 1 and 3), and one sub-family is in the same clade with fly orthologs (*Figure 4A*). Among these 4 clades, only the *fmo* genes in clade one were found significantly enriched in both of the pigment cell states. *Fmo5-1, fmo3* and *fmo2-2* (previously named 1,2,3) were all expressed in the mitotic and differentiated pigment cell clusters, but not in the mesodermal cluster 12 (violin plots in *Figure 4—figure supplement 1*; *Figure 4L–M*). We next investigated the co-expression of clade one *fmos* with *gcm* and *pks1* by double in situ hybridization (*Figure 4B–I'*). We found that *fmo5-1* and 3 were exclusively expressed in pigment cells, while *fmo2-2* is expressed also in cells that are *gcm*-negative (*Figure 4D–D'*). *Fmo2* was not detected in the ectoderm single cell transcriptomes, but we analyzed its spatial expression since it was at the base of clade 1 (*Figure 4A*). In addition to the pigment cells, *fmo2* was expressed at low level in a few cells of the blastocoel that are *gcm/pks1* –. They are in tight apposition to the ectoderm (*Figure 4E,E' and I,I'*). Overall, by scRNA-seq and FISH we found that all *fmos* from clade one were enriched in pigment cells. In a different sea urchin species (*Hemicentrotus pulcherrimus*), knocking out the gene function of *fmo3* resulted in a change in pigmentation of the adult, distinct from the albino phenotype of the *pks1* gene knockout (*Wessel et al., 2020*). Perhaps these Fmo family members uniquely have enzymatic activities for polyketide metabolites, so that in the pigment cell the polyketide derived from Pks1 is converted to the intensely deep purple pigment.

We next determined the expression of *fmos* that were not found in the scRNAseq transcriptome. We used the available data of *S. purpuratus* quantitative RNA expression through development (http://www.spbase.org:3838/quantdev/) as an additional tool to determine the temporal expression of representative genes from the 4 clades of *fmos*. Clades 2, 3 and 4 *fmos* were expressed at low levels during embryonic development, with less than 1000 transcripts per embryo (*Figure 5A,B*). Only *fmo2-3* from clade three was expressed between 1000 and 2000 transcripts/embryo. All transcripts of clade one *fmos* were instead above 10,000 transcripts/embryo, reflecting their high abundance in the embryos. Only *fmo2*, located at the base of clade one in the phylogenetic tree, was not abundant in the embryo and had less than 250 transcripts/embryo (*Figure 5A,B*). We investigated further the spatial expression of representative genes for the other three *fmo* clades that are poorly expressed at embryonic stages. *Fmo1, fmo2-3* and *fmo5* are not exclusively expressed in the pigment cells, and FISH analysis showed low expression levels in the whole embryo (*Figure 5—figure supplement 1*). Our experiments show that only clade one *fmos* are highly expressed in embryos (in agreement with our FISH and transcriptomic analyses), while the other clades of *fmos* are poorly or not detectable at all in embryos.

Since our results showed that many *fmos* were not expressed during development, we analyzed their temporal expression in adult pigmented structures (spines and tube feet) and in the adult immune system (coelomocytes) by performing quantitative PCR in these tissues (*Figure 5C–E*). We first documented that spines and tube feet have pigmented cells by the criteria of deep red color and immunoreactivity to a conserved pigment cell marker recognized by the antibody Sp1

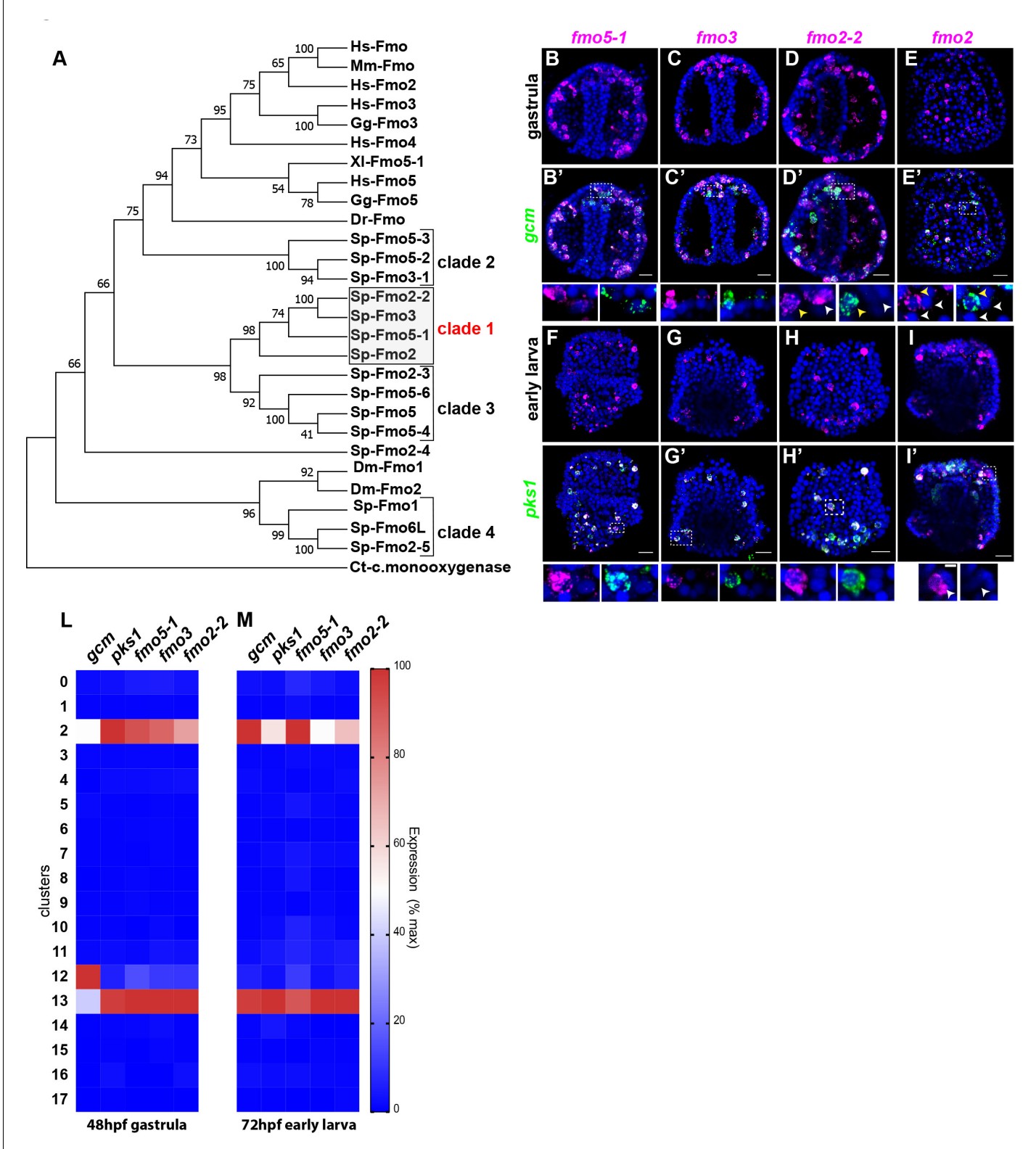

**Figure 4.** A unique group of *fmo* genes is specifically expressed in pigment cells. (**A**) Maximum Likelihood tree showing the relationship of sea urchin flavin-dependent monooxygenases (FMOs) to other FMO proteins. Branches corresponding to partitions reproduced in less than 50% bootstrap replicates (500 replicates) are collapsed. Baeyer–Villiger monooxygenase was used as an outgroup. The same phylogeny was also supported by a maximum parsimony tree. (**B–I'**) Double FISH of fmos from clade one with *gcm* or *pks1*. In the insets, yellow arrowheads indicate co-expression in the

*Figure 4 continued on next page*

Figure 4 continued

pigment cells, white arrowheads indicate cells that are not pigment cells. Dashed line boxes indicate cells magnified below each picture. Nuclei are depicted in blue (DAPI). Scale bars 50 µm in whole embryo images and 20 µm in magnifications. All images are stacks of merged confocal Z sections. (L–M) single cell data show that the fmos of clade one are found in clusters 2 and 13.

The online version of this article includes the following figure supplement(s) for figure 4:

**Figure supplement 1.** Distribution of *fmos* in the single cell transcriptome shows high *fmo* gene expression in cluster 2, 12 and 13.

(*Gibson and Burke, 1985*; *Figure 5C,D*). We found that at least one gene of clades 2, 3 and 4 was expressed in the spines, while *fmos* from clade one were lacking (*Figure 5C*). Tube feet were not a site of significant *fmo* expression, only two of the *fmos* were detected (*Figure 5D*). Coelomocytes, however, expressed *fmos* from clade 1, but not the other *fmos* (*Figure 5E*). These cells are diverse in morphology and putative function, of which the red spherule cells are known to produce echinochrome and spinochrome (*Coates et al., 2018*; *Hira et al., 2020*). In the pigment biosynthesis pathway, the enzyme *pks1* is upstream of the *fmos* in the biosynthetic pathway of echinochrome. We found that *pks1* was highly expressed in the adult immune cells, but expressed only at low levels in spines and tube feet. We interpret these results as *fmos* might function independently from *pks1*, and that low levels of polyketide synthase gene activity are sufficient for maintenance levels of polyketide synthesis and pigmentation, especially in immune-quiescent animals (not challenged by bacteria). Taken together, these results suggested that of the four *fmo* families, only clade 1 was found specifically in pigment cells of the embryo and in the adult red spherule cells, two cell-types with known immune functions. Within this clade, *fmo2-2* and *fmo2* are expressed also in embryonic cells that do not contain pigments, pointing these to a distinct oxidizing function. Other *fmos* are expressed in the spines (clades 2, 3 and 4) where *pks1 is* expressed at low levels, suggesting a role for non-clade 1 *fmos* independent of pigment biosynthesis and from *gcm* and *pks1*. Overall, we conclude that the many pigmented cell populations in the larvae and adults of *S. purpuratus* have distinct gene profiles.

## Pigment cell dependence on *gcm* function

Having defined that *gcm* expression is associated with the expression of pigment cell genes, we tested whether pigment cell gene expression was dependent on *gcm*. Gastrulae in which *gcm* is knocked down have a unique phenotype with no pigmentation and low levels of *pks1*, *sult1c2* and *fmo3* (*Ransick and Davidson, 2006*). Even if some indirect evidence exists (*Ransick and Davidson, 2006*) a direct test of pigment cells presence/absence in gcm KD or null embryos is still missing. To understand whether in *gcm* KD embryos pigment cells are 'pigmentless' because of a lack of *pks1* and *fmo3*, or whether there are fewer pigment cells, we tested *gcm* role in pigment cells early specification and pigment production. We performed a single cell differential gene analysis of 48hpf embryos compared to control embryos to determine the fate of the pigment cell clusters when *gcm* is knocked down. We dissociated control and *gcm* KD gastrulae without enriching for the ectoderm to ensure that all cells were captured for the analysis. Therefore, these tSNE plots (*Figure 6A*) revealed information different than the scRNAseq of ectoderm-enriched wild-type embryos (*Figure 1B*). The tSNE plots of the single cell transcriptome of control gastrula show a main group of ectodermal clusters (ciliary band neurons, serotonergic neurons, apical and aboral ectoderm), and clusters for skeleton, coelomic pouches, foregut and mid-gut (*Figure 6A*). Cells from gcm-KD embryos were very sensitive to dissociation and more susceptible to lysis than normal embryos. Because of the low number of sequenced cells, and the consequent lack of many ectodermal cell types, pigment cells were represented by a single cluster (cluster 6). The *gcm* KD experiment resulted in the same tSNE cluster profile as with the control MO. Since fewer cells were sequenced in the *gcm* MO experiment as a result of greater sensitivity to dissociation, we normalized the cells in each cluster for the total sequenced cells in each experimental condition. While 40% of the clusters did not change size (*Figure 6—figure supplement 1*), the pigment cell cluster population dramatically decreased by a factor of 7 with *gcm* knocked down by the morpholino (*Figure 6B*). We interpret this result as *gcm* is responsible to specify pigment cells, and not just for driving expression of pigment cell genes.

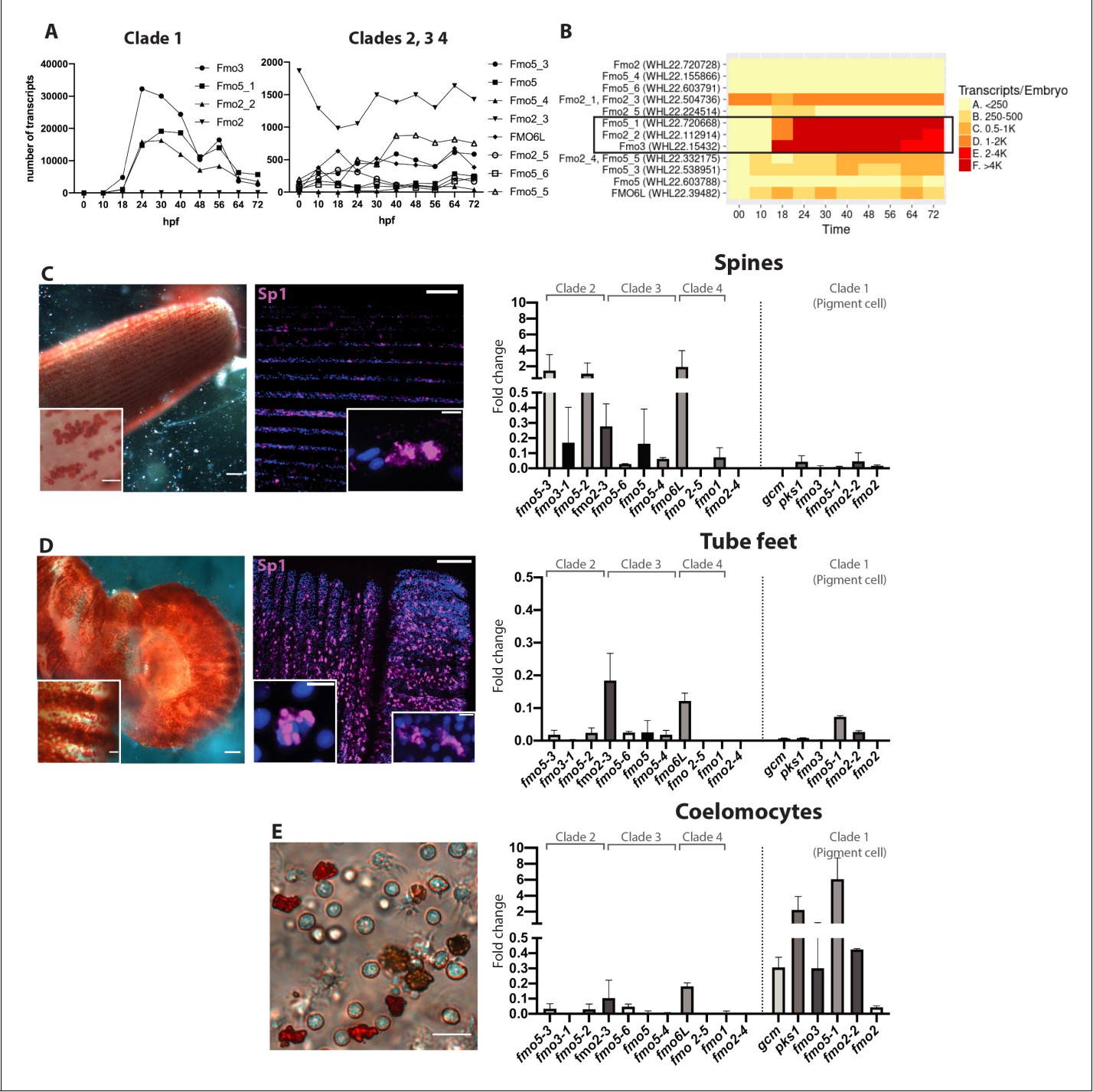

**Figure 5.** Fmos from clade one are expressed in embryonic pigment cells and in adult coelomocytes. (A, B) Transcripts abundance of the four fmo clades in the embryo, 0 to 72 hr post fertilization (hpf). This graph was made using the online resource http://www.spbase.org:3838/quantdev/. All *fmos* in clade 1 (*fmo5-1, fmo2-2, fmo3*, squared box) and *fmo2-3* from clade three are expressed in the embryo. A and B represent the same data. (C, D, E) Spine, tube feet and coelomocytes bright field images show red pigmented cells. Scale bars are 100 µm, magnification scale bars 20 µm. Confocal images of spines and tube feet labeled with the pigment cell marker Sp1, showing that adult structures have common features with embryonic pigment cells. Nuclei are in blue (DAPI); scale bar 100 µm and 5 µm for insets. qPCR of fmos in the adult spines (C), tube feet (D) and coelomocytes (E) show that clade one is expressed in coelomocytes. *Gcm* and *pks1* are expressed in the coelomocytes, but not in spines or tube feet. All experiments were repeated in three biological replicates. Fold change = 2(-Dct). FISH of genes not in clade one is in *Figure 5—figure supplement 1*.

The online version of this article includes the following figure supplement(s) for figure 5:

**Figure supplement 1.** Spatial expression of other *fmo* genes that are not present in clade 1.

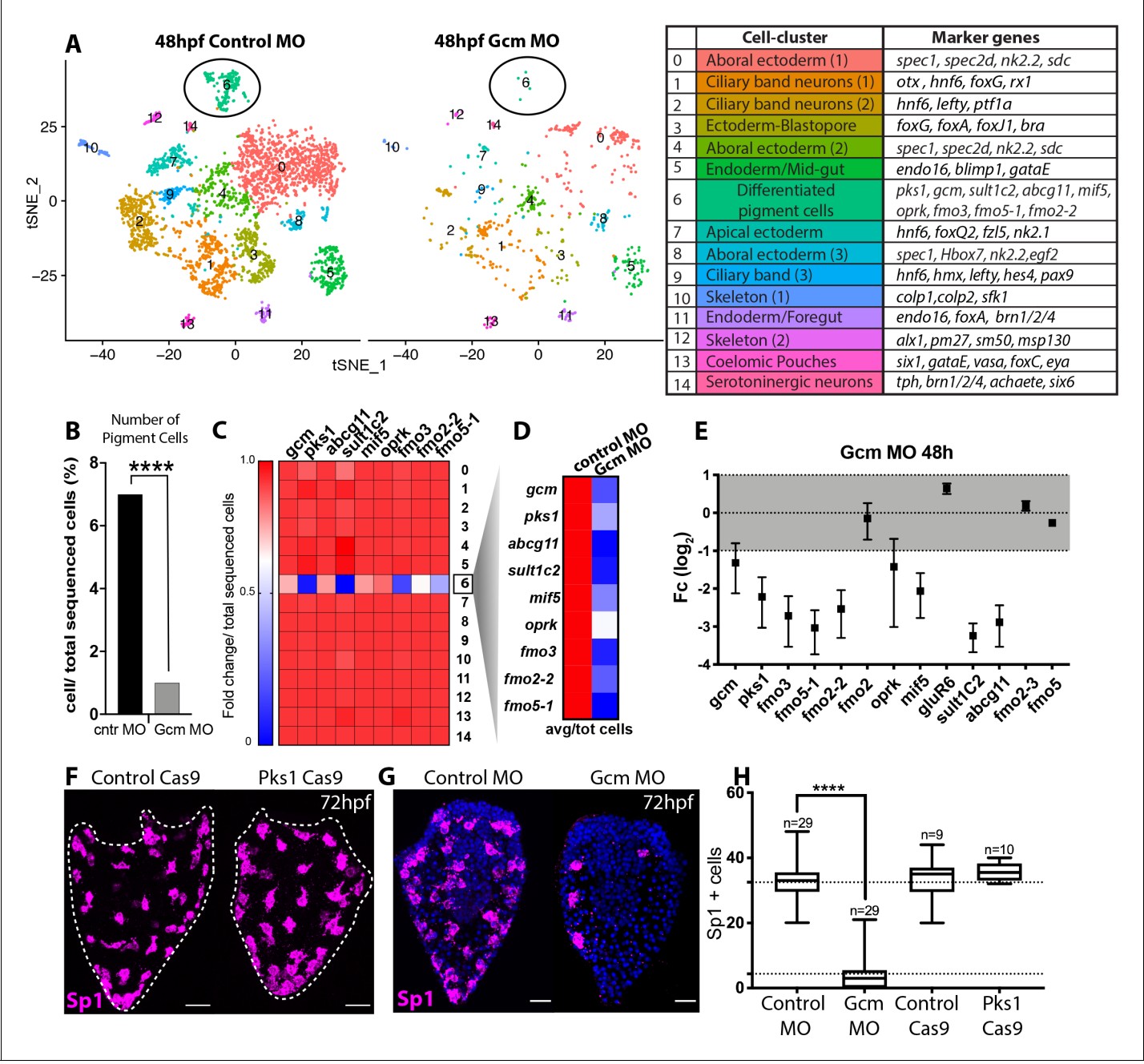

**Figure 6.** Gcm controls pigment cell specification. (**A**) Global visualization of single cell profiled in control MO and gcm MO gastrulae. tSNE plots show that in controls and morphants there is no change in general cluster organization. Pigment cell cluster is number 6, green. The table summarizes the main cell-types and relative marker genes, colors represent clusters in the tSNE plots. In the 48hpf control morpholino injected sample, 3402 single cells were captured for downstream analysis, sequenced at a level of 56,550 reads per cell. The 48hpf gcm morpholino sample includes 643 single cells with an average of 379,895 reads per cell. The two samples were integrated to identify conserved cell types and cluster marker genes using Seurat. Feature and violin plot of pigment cell genes in controls and gcm depleted embryos are shown in *Figure 6—figure supplement 2* and *Figure 6—figure supplement 3*. (**B**) Percentage of Pigment Cells relative to total sequenced cells in control MO versus *gcm* MO experiments, two-sided Chi-square and Fisher's exact test gave p<0.001 (****); Chi-square df 31,35, 1, z 5.599. A graph with percentage of cells relative to total sequenced cells for each cluster is shown in *Figure 6—figure supplement 1*. (**C**) Expression of pigment genes in gcm MO (Fold change, Fc) compared to control MO normalized by number of total sequenced cells in each experiment (average LogFc*number of cells in each cluster/total sequenced cells in each experiment). (**D**) Expression levels of pigment genes normalized for the total sequenced cells shows that there is an overall decrease in gene expression in the pigment gene clusters. (**E**) qPCR of pigment genes in gcm MO gastrulae. Experiments were performed in three biological replicates (error bars). (**F**) Sp1 Ab staining of 72hpf larvae depleted of *pks1* by CRISPR Cas9 shows that distribution of pigment cells does not change. A qPCR of pigment genes in control and *pks1* depleted embryos is shown in *Figure 6—figure supplement 4*. (**G**) Sp1 Ab staining in *gcm* MO early larva (72hpf). Nuclei are stained

*Figure 6 continued on next page*

*Figure 6 continued*

with DAPI. (**H**) Number of Sp1 positive cells decreases in gcm MO compared to controls (n = number of larvae, two-tailed student t-test p<0.001 (****) corresponds to p=1,44704E-17; 95% confidence interval of −31,07 to −25,21), (mean for control MO is 35,51; mean for *gcm* MO is 4.37), and it is unchanged in *pks1* Cas9 (two-tailed student t-test p value 0,4191; 95% confidence interval of −3,326 to 7,393; mean of control Cas9 is 33,67, mean of pks1 Cas9 is 35,70). Additional file in *Figure 6—source data 1*. Images are confocal Z-projections, scale bar 50 μm.

The online version of this article includes the following source data and figure supplement(s) for figure 6:

**Source data 1.** Number of Sp1+ cells in *gcm* MO, control MO, pks1 Cas9 and Cas9 controls.
**Figure supplement 1.** Number of cells in each cluster in control embryos versus embryos depleted of Gcm.
**Figure supplement 1—source data 1.** Distribution of pigment genes in single cell transcriptomes of control and gcm depleted embryos.
**Figure supplement 2.** Distribution of pigment genes in single cell transcriptomes of control and gcm depleted embryos.
**Figure supplement 3.** Distribution of pigment genes in single cell transcriptomes of control and gcm depleted embryos.
**Figure supplement 4.** Gene expression of candidate genes in pks1- larvae is indistinguishable from controls.

We next examined the overall expression level of pigment cell genes following *gcm* knockdown, and found that the expression of most of the tested genes decreased in cell state cluster 6 (*Figure 6C* shows fold change; feature plots and violin plots in *Figure 6—figure supplement 2* and *Figure 6—figure supplement 3*), while *gcm* expression remained unchanged. In *gcm* morphants, the expression of pigment genes in the embryo decreased because there were significantly fewer pigment cells (*Figure 6D* shows gene expression in controls versus morphants). To focus the analysis on *gcm* targets, we tested by qPCR expression level changes of other *gcm* targets (*Figure 6E*). The expression of *gluR6* and *fmo2*, which were expressed in more cell-states than the *pks1*+ cells, was unchanged. *Fmo2-2* from clade1, which was not detectable in the single cell experiment, was down-regulated in the *gcm* morphant. Genes like *fmo*2-3 and *fmo*5, not found in the pigment cell cluster, did not change expression in gcm morphant embryos (*Figure 6E*). In contrast, larvae of cas9-mediated knockout of *pks1* were indistinguishable from control larvae when stained for Sp1 antibody (*Figure 6F*). The *pks1* - KO is used here to test homeostasis in the pigment cells and whether Pks1 and/or pigment may have any function, directly or indirectly, in regulating gene expression. It has been previously shown that embryos lacking Pks1 still have pigment cells (*Calestani et al., 2003*). Pigment cells stained with the Sp1 antibody show that not only the number, but also the distribution of pigment cells does not change in P*ks*1 depleted larvae. Gene expression of candidate genes in pks1- larvae was also indistinguishable from controls (*Figure 6—figure supplement 4*). We conclude that pigmentation does not have a role in positioning these cells within the larvae, in regulating numbers or density of the pigment cells, or change overtly the physiology of the cells as this would have been revealed by the cellular phenotype or steady state levels of relevant candidate mRNAs.

Having shown that *gcm* defines the number and gene expression of pigment cells, we tested in vivo the localization of surviving pigment cells in the absence of *gcm*. To this aim, we stained surviving pigment cells with Sp1 in early larvae, when pigment cells reach their final position. In agreement with the single cell data, we find fewer pigment cells in the *gcm* MO larvae compared to controls, and their localization appears to be randomly dispersed throughout the aboral ectoderm (*Figure 6G*). Taken together our experiments show that *gcm* knock down compromises the number of pigment cells .

## Discussion

### *Gcm* as a regulator of pigment

Pigments are compounds that selectively absorb a certain wavelength, and their production relies on a complex biosynthetic pathway that, for most plants and animals, involves ill-defined pathways. The cells that make the pigment are diverse and distinct in their biology, and the echinoderm pigments themselves also likely serve diverse functions. For instance, spinochromes and other pigments have been identified in adult of *Echinometra mathaei*, a species of sea urchins with many color variations between adults of the species, and pigmentation in this animal is correlated with fitness (*Brasseur et al., 2018*). It has also been proposed that the water-soluble pigment Spinochrome E can penetrate into the egg cytoplasm of the sand dollar *Scaphechinus mirabilis* and be used as a source of hydrogen peroxide in the embryo (*Brasseur et al., 2018*; *Drozdov et al., 2017*). Here we

use a combination of RNA single cell sequencing and fluorescent in situ hybridization to take a deep look at the genes responsible for this incredible variety of colors, and we show that pigment cells within an organism are highly diverse (*Figure 7*).

*Gcm* has been shown to be expressed first in the Veg2 lineage during early blastula and to be later restricted to pigment cells (*Ransick and Davidson, 2006*; *Ransick and Davidson, 2012*; *Ransick et al., 2002*). There was some preliminary evidence that *gcm* might be expressed in other cell types, that is the coelomic pouches, later in larval development (*Ransick and Davidson, 2012*). Here, by integrating the *gcm* positive cells from the ectodermal scRNAseq with a time course over eight embryonic stages, we found that *gcm* was expressed broadly and that its expression was dynamic. Since Notch signaling is activated not at the same time in the ring of mesodermal cells surrounding primary mesenchyme cells (*Sherwood and McClay, 1997*; *Sweet et al., 1999*), it is possible that *gcm* and downstream genes are activated at slightly different times, one cell after the other. Our conclusion is that *gcm* is not essential for all these clusters, but it has only a transient expression regulated at the transcriptional and/or post-transcriptional level. This suggests either a housekeeping role for *gcm* that was underestimated, or a role in initiating transcription of certain genes, whose expression will be maintained by other regulatory factors following possibly an additional environmental cue or cell signaling. Last, of the three *gcm*+ clusters, cluster 12 defines mesodermal pigment cell progenitors (*six1* and *eya* markers), determined but not differentiated yet. This dynamic *gcm* expression would have been more difficult to establish without single cell analysis, since it was not detectable so far using other techniques.

It is known that the downstream effects of *gcm* loss of function is the lack of pigment cell precursors migration during gastrula and lack of pigment synthesis (*Ransick and Davidson, 2006*). Here we show that *gcm* is responsible for specifying pigment cells, not just driving expression of pigment cell genes. ScRNA-seq of *gcm* morphants shows that in the absence of gcm the number of pigment cells decreases, and the expression of pigment gene markers is reduced. Moreover, immunostaining

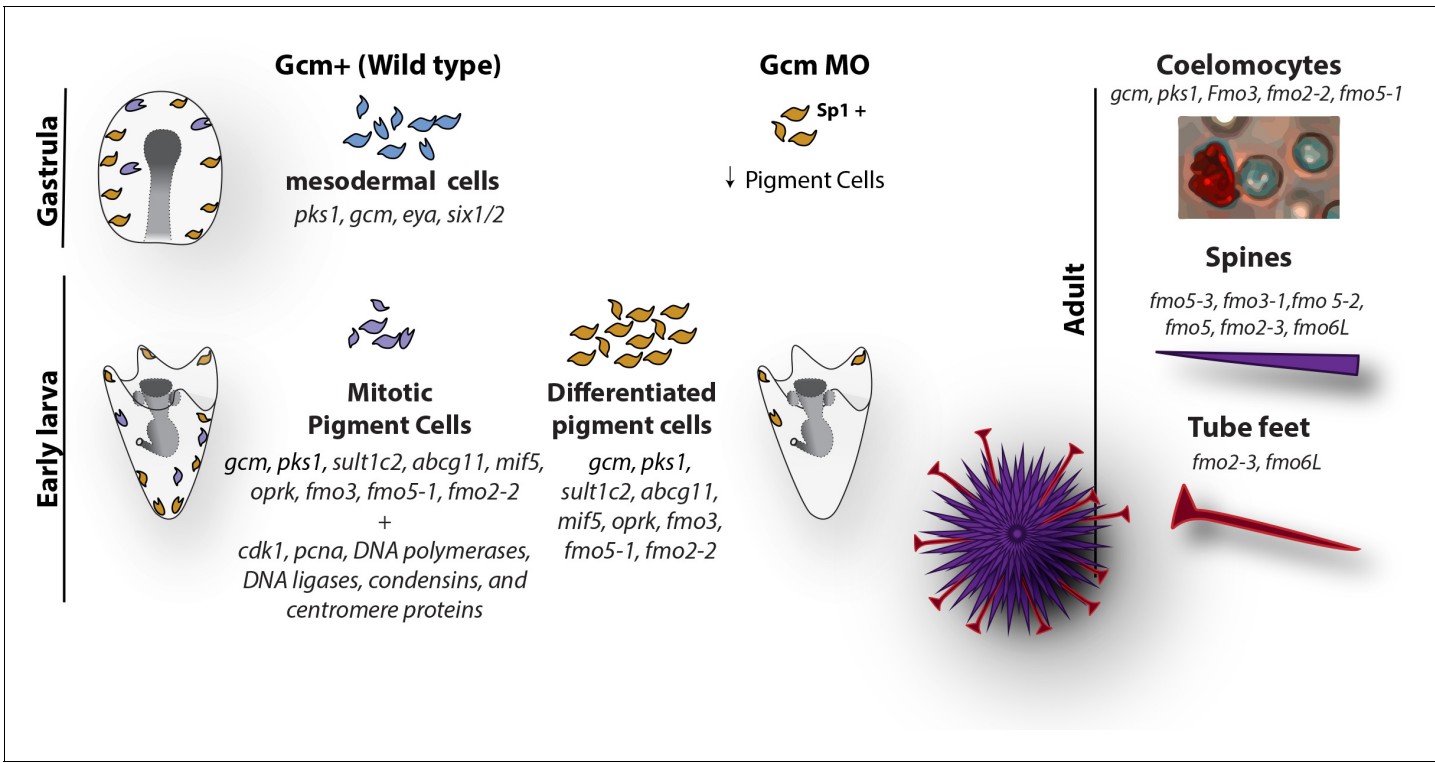

**Figure 7.** Diagram summarizing the fate of pigment cells in wild type and gcm MO embryos. In wild type embryos there are 3 subpopulations of cells expressing *gcm*: mesodermal cells not yet differentiated, differentiated pigment cells and mitotic pigment cells. Colors are derived from the clusters of *Figure 1*. Larvae have both differentiated and mitotic pigment cells. In embryos where *gcm* is depleted there are fewer pigment cells. Adult sea urchins express different levels of *fmo* genes in their pigmented structures. *Fmos* that are expressed in the embryonic pigment cells are also expressed in the adult red spherule cells, both groups of cells having immune functions.

with the Sp1 antibody found that a small population of pigment cells remain when Gcm is depleted, but we can't exclude the existence of *gcm*-independent compensatory mechanisms. Our results suggest that despite *gcm* being necessary to specify most pigment cells, *gcm*-independent mechanisms appear to activate pigment cell identity when *gcm* activity is depleted. We cannot, however, exclude the possibility that residual *gcm* expression somehow fully activates a small population of precursor pigment cells.

A previous transcriptome study identified pigment cell genes by enriching the sample for late gastrula *gcm*-GFP expressing cells (*Barsi et al., 2015*). To evaluate our single cell data in this comparative light, we searched for the pigment cell markers described in our study in the list of genes previously reported, and found that all of the candidates we identified were enriched in the above gcm+ transcriptome. Some of the pigment markers we found were also expressed in other cell states. For example, *oprk*, that in other organisms modulates feeding and dopamine-induced behavior (*Crawley, 1991*) and the glutamate receptor *gluR6*, were expressed in pigment cells and also in the ciliary band neurons (cluster 16). In particular, *gluR6* is expressed in a cluster of neurons located in the blastocoel basal to the ciliary band that resemble the localization of the lateral ganglion cells (*Bisgrove and Burke, 1987*; *Perillo et al., 2018*). We interpret the neuronal expression as new gene expression in those cells, and not a cell fate transition of the pigment cells since pigment cells are differentiated in larvae. The fact that pigment cells were expressing these neuronal receptors and others (this study and [*Barsi et al., 2015*]) suggests that they might also have a sensory function possibly coordinated with their immune response function. Our study provides new insights into the expression and function of *gcm*, a transcriptional factor that we found dynamically expressed in several embryonic clusters, and that controls the different states of pigment cells and possibly their different functions.

## Do pigment cells maintain a stem state?

How the pigment cells are regulated through larval development and into adulthood is not clear. Do all the various pigment cells of the adult, including in the spines, tube feet, and coelomocytes, come from a core stem cell population? Or do they each have their own developmental lineage from a unipotential precursor? Each of these populations, as shown here, appear to have different transcript profiles and functions. What is clear is that the numbers of pigment cells expand if larvae are immune challenged (*Kiselev et al., 2013*; *Ageenko et al., 2014*; *Wessel et al., 2020*). In our scRNA-seq of ectodermal cells we found two distinct populations of pigment cells, one of which is enriched in mitotic markers, suggesting the possibility of a stem cohort of dividing cells. We favor the concept of a stem population for this lineage because the transcript profile is distinct in the mitotic cohort and those cells are resistant to the *gcm* knockdown. Perhaps Gcm is essential for the lineage expansion, but not for the stem precursors.

The molecular analysis does suggest distinct populations, or at least different cell states, of pigment cells. By labeling mitotic cells with a pulse of EdU and pigment cells with the marker Sp1, we visualized the mitotic pigment cells in vivo. These cells do not seem to belong to a specific embryonic stem niche, but are distributed throughout the embryo, suggesting that a portion of all pigment cells undergo mitosis to increase the pool. The presence of mitotic pigment cells at the early larva stage also suggests that these cells retain a very active cell cycle while migrating. With such essential functionality likely in the innate immune defense, it may be expected that a stem cell-type population may be responsible for homeostatic functions in pigment cell populations. Pigment cells from embryos of the sand dollar *S. mirabilis* and in the sea urchin *S. intermedius* can be induced to differentiate when cultured in the adult coelomic fluid in vitro, suggesting that genes involved in pigment synthesis in the adult also regulates pigment cells of the larva (*Ageenko et al., 2016*). Our finding that adult coelomocytes from *S. purpuratus* express *gcm*, *pks1* and group1 *fmos* suggests that pigmented cells from the adult cells share biosynthetic and cell fate pathways.

We pose two hypotheses to explain the finding of a mitotic pigment cell cluster: either there is a unique stem cell population of pigment cells, or all pigment cells have the ability to divide. In both cases, a follow up question is whether they divide upon a signal, and what this signal might be (i.e. coelom infection, arm growth, ectodermal injury). It has been suggested that in adult sea urchin a population of pigment stem cells is set aside in the pre-metamorphosis larva to serve as a reservoir for the adult cells (*Wessel et al., 2020*). This is more than initially activating the program for these cells to develop, but also to maintain and replace these cells in the adult. We can rule out pigment

presence as an indicator of homeostasis since the *pks1* knock out larvae had similar pigment cells distribution as wild-type larvae. Thus, a stem population for these cells may be key for adult survival and the single cell results herein may point to this population. We propose that the hereby-identified embryonic mitotic cluster might be the cells that are selected in early embryos to replenish the larvae and/or adult stages. However, as we learned here, pigmented cells in the diverse tissues of an adult are pigmented cells in distinct states. Either the presumed stem cluster is capable of diversifying, or new pigment lineages form later in development. An alternative hypothesis is that the pigment cell mitotic cluster serves in the embryo as a reservoir to replenish blastocoelar cell-types. We propose that the mitotic pigment cell cluster might represent a group of cells that retain the developmental potential that allows these cells to convert to other blastocoelar cells, or to the highly migratory pigment cells of the larvae. Overall our interpretation of mitotic and post-mitotic pigment cell clusters is compatible with cells in the latter cluster having immune function, whereas the mitotic pigment cells population may be a stem/dividing population. Indelible lineage labeling will be important to test these hypotheses.

## A unique family of *fmos* is expressed in cells with immune functions

Spines, tube feet, cells in coelomic fluid and larvae of sea urchins express different types of flavin containing monooxygenases (*fmo*) enzymes that contribute to the pigment biosynthesis. In adult spines and tube feet under control conditions (not immune challenged) *pks1* is not transcribed, or is so at very low levels. Two scenarios are possible to explain this observation: either cells could have a level of enzyme (protein) sufficient to synthesize some pigment or in the absence of degranulation of pigment in response to infections or injuries, the synthesis of pigment is halted. In the case of cells with predicted similar functions, for example coelomic-pigmented cells (the red spherule cells), versus larval-pigmented cells, the composition of pigment synthesizing enzymes and other gene expression is instead similar. Cells with immune functions (adult coelomocytes and larval pigment cells) always express the same group of *fmos*, together with the enzyme *pks1* and the transcriptional factor *gcm*. Both cells produce the pigment echinochrome A, a molecule with antimicrobial properties (*Brasseur et al., 2018*; *Coates et al., 2018*; *Gerardi et al., 1990*; *Johnson, 1969*; *Lebedev et al., 2005*; *Perry and Epel, 1981*), suggesting that *gcm*, *pks1*, and group1 *fmos* work together to make this unique pigment, but might not interact to make other echinoderm pigments. These results may reveal that distinct biochemical pathways lead to the same chemical outcome, or that pigmentation is just one of multiple different functions in what we routinely refer to as a single pigment cell.

The revelation of the *fmo* family in sea urchins is striking. *Fmos* are broadly expressed in yeast, plants and animals where their main role is detoxification (*Dolphin et al., 1996*; *Janmohamed et al., 2001*; *Phillips et al., 1995*; *Phillips and Shephard, 2017*). Seeing diverse expressions throughout the sea urchin tissues suggests more than 'just' a metabolic or detoxifying function for these family members. Fmo enzymes catalyze the oxidation of different types of substrates that are involved in different types of biological processes across phyla: detoxification, metabolite activation, and in the case of the sea urchin, pigmentation and maybe more. In the sea urchin, *S. purpuratus*, one cohort of *fmos* (Group 4, that includes *fmo1, 2–5* and *6*) groups together with the fly orthologs. Flies have two *fmos,* one with housekeeping functions and another involved in development (*Scharf et al., 2004*). *Fmos* from clade two are instead grouped with the human Fmos. These *fmos* were not detectable in embryogenesis, but rather in the spines. Human *fmos* are expressed in liver, lung, kidney and skin (*Peters, 1985*; *Schlaich, 2007*), cannot be induced or inhibited, and although being involved with drug metabolism, their physiological role is poorly understood (*Scharf et al., 2004*). The most studied FMO is the human Fmo3, an enzyme that is constitutively active in the adult human liver where it participates in the oxidation of drugs and xenobiotics, and mutations in Fmo3 contribute to the disease trimethylaminuria (*Cashman and Zhang, 2002*; *Dolphin et al., 1996*; *Krueger and Williams, 2005*). We find no expression of the *fmos* in clade 2 (the sea urchin group closest to vertebrates) in embryos, nor in the coelomocytes, as it is only expressed in spines. Sea urchin spines are covered by a thin epithelium (*Peters, 1985*) that is populated by pigmented cells that we show here are marked with the embryonic pigment cell marker Sp1, most likely being red spherule cells. We suggest that the *fmo* clade two role might be connected to human Fmo function in the skin, where it might detoxify xenobiotics to which the spines are exposed in the seawater. The other fmo clades are also expressed in the spines, and further studies will need to determine the expression and function of these genes in other adult structures.

It is still not understood why the *fmo* family is so expanded in this sea urchin. Only 1 *fmo* gene is present in yeast, whereas many flowering plants have 26 (*Schlaich, 2007*), two in flies (*Scharf et al., 2004*), six in humans (*Dolphin et al., 1996*) and at least 12 in sea urchin that are not pseudogenes, since each are expressed at some time and place in the animal. A hypothesis to explain this expansion is that such a variety of enzymes is necessary for the detoxification of a wide range of xenobiotics present in diverse ecosystems, especially benthic niches. We predict that with Cas9 targeting mechanisms of *fmos* followed by pigment MS/MS analysis will enable definitive Fmo functionality to expose this enigmatic family of enzymes. Furthermore, single cell RNA-seq of adult tissues will likely reveal the cell-types that express these genes.

In conclusion, by leveraging two distinctly derived RNA-seq datasets, we were able to create confident profiles of genes relevant to pigment cell function. We discovered that pigment cells conserve a mitotic cell population that might represent a stem population for embryonic and adult immune cells. We also realize how important in situ FISH and single-cell RNA-seq is for confidence in conclusions. Bulk sequencing, even isolated populations of cells, brings many caveats of interpretation which likely shift upon testing with scRNA-seq and double-FISH.

# Materials and methods

## Key resources table

| Reagent type (species) or resource | Designation | Source or reference | Identifiers | Additional information |
|---|---|---|---|---|
| Gene (*Strongylocentrotus purpuratus*) | *gcm* | Echinobase | SPU_006462 | |
| Gene (*Strongylocentrotus purpuratus*) | *pks1* | Echinobase | SPU_002895 | |
| Gene (*Strongylocentrotus purpuratus*) | *fmo2* | Echinobase | SPU_002963 | |
| Gene (*Strongylocentrotus purpuratus*) | *fmo3* | Echinobase | SPU_017374 | |
| Gene (*Strongylocentrotus purpuratus*) | *fmo2-2* | Echinobase | SPU_014947 | |
| Gene (*Strongylocentrotus purpuratus*) | *fmo5-1* | Echinobase | SPU_012348 | |
| Gene (*Strongylocentrotus purpuratus*) | *sult1c2* | Echinobase | SPU_006187 | |
| Gene (*Strongylocentrotus purpuratus*) | *oprk1c* | Echinobase | SPU_000719 | |
| Gene (*Strongylocentrotus purpuratus*) | *mif5* | Echinobase | SPU_020036 | |
| Gene (*Strongylocentrotus purpuratus*) | *glur6* | Echinobase | SPU_028455 | |
| Gene (*Strongylocentrotus purpuratus*) | *abcg11* | Echinobase | SPU_020849 | |
| Antibody | anti-Sp1 (mouse monoclonal) | DSHB | Cat# Sp1, RRID:SCR_013527 | IF(1:50) |
| Sequenced-based reagent | control MO; control morpholino | Gene Tools | | 5'-GCTTTGGAGTAACCTTCTGCACCAT-3' (0.5 mM) |

*Continued on next page*

*Continued*

| Reagent type (species) or resource | Designation | Source or reference | Identifiers | Additional information |
|---|---|---|---|---|
| Sequenced-based reagent | gcm MO; gcm morpholino | Gene Tools | | 5'-GCTTTGGAGTAACCTTCTGCACCAT-3' (0.5 mM) |
| Software, algorithm | Image J | Image J | RRID:SCR_003070 | |
| Software, algorithm | Seurat v 3.1.4 | SEURAT | SEURAT, RRID:SCR_007322 | |
| Other | DAPI stain | Invitrogen | D1306 | (1 μg/mL) |

## Animals

Adult *Strongylocentrotus purpuratus* were obtained from Josh Ross (info@scbiomarine.com) off the California coast and kept in artificial seawater at 16°C. Gametes were obtained by shaking adult sea urchins or by intracoelomic injection of 0.5M KCl. Eggs were fertilized in the presence of 1 mM 3-amino-triazol (3-AT) (Sigma, Cat. #:A8056). Embryos were cultured at 16°C in filtered seawater from the Marine Biological Laboratory (MBL).

## RNA whole mount in situ hybridization

For fluorescent whole mount in situ hybridization (FISH), we followed the protocol outlined in *Cole and Arnone, 2009*; *Perillo et al., 2016*. Signal was developed with fluorophore-conjugated tyramide (Perkin Elmer, Cat. #:NEL752001KT; RRID:AB_2572409). Labeled probes were transcribed from linearized DNA using digoxigenin-11-UTP (Sigma Aldrich, Cat. #:11277073910), or labeled with DNP (Mirus Bio, Cat. #:MIR3825) following kit instructions. Probes were synthesized using primers in *Supplementary file 3*. 20 to 40 samples were imaged with a Zeiss 800 confocal microscope from the Brown University Leduc Bioimaging Core Facility (RRID:SCR_017861).

## Immunochemistry and EdU pulse labeling

Larvae and adult tissues were fixed in 4% paraformaldehyde (PFA) in filtered seawater (FSW) for 15 min at room temperature, washed multiple times in phosphate-buffered saline with 0.1% Tween-20 (PBST), and incubated overnight at 4∘C with the Sp1 antibody 1:50 (DSHB; RRID:SCR_013527) in 1 mg/ml Bovine Serum Albumin (BSA) and 4% sheep serum in PBST. Samples were then washed three times with PBST and incubated for 2 hr at room temperature with the secondary anti-mouse antibody conjugated to Alexa 488 (Life Technologies; Cat#:A-21121, RRID:AB_2535764) diluted 1/500 in 1 mg/ml BSA in PBST. 5-ethynyl-2-deoxyuridine (EdU) pulse labeling of the pigment cells was performed with the Click-IT EdU imaging kit (Life Technologies; Cat#:C10340). Embryos and larvae were soaked in 10 μM EdU in FSW for 30 min (considering a 20 min cell cycle), washed five times with FSW and fixed in 4% PFA/FSW. Following EdU detection with fluorescent azide (according to the manufacturer's instructions), samples were stained with the Sp1 antibody as described above. 20 to 40 larvae were stained for imaging. Samples were mounted for imaging with an Olympus FV3000 Confocal Microscope (RRID:SCR_017015) equipped with high sensitivity GaAsP detectors managed by cellSens software (RRID:SCR_016238) from the Brown University Leduc Bioimaging Core Facility (RRID:SCR_017861). Raw files were analyzed and figures were made using the software Image J (RRID:SCR_003070).

## Perturbation experiments with MO injection and CRISPR Cas9

Translation-blocking antisense morpholino (MO) against Gcm (SPU_006462; MO synthesized by Gene-Tools; sequence 5'-GCTTTGGAGTAACCTTCTGCACCAT-3') was used at a concentration of 0.5 mM. MOs were injected with 20% glycerol and 10,000 MW fluorescent dextran (injection solution). Eggs were dejelled by washing in pH4.0 seawater. For each experiment, around 600 zygotes were injected with ~2–4 pl of oligonucleotide injection solution by constant pressure injection in with 1 mM 3-AT (Sigma). For each condition, only 300 of these injected embryos with the expected phenotypes were dissociated at 48hpf and used for single cell RNA seq. The *gcm* morphant has a unique phenotype with a straight gut and no pigmentation. As a negative control, fertilized eggs were injected with a MASO sequence irrelevant to *S. purpuratus* (the MO to Foxy3 of *Patiria miniata*,

and is referred to as the control morpholino [5'-TGCGATTAGAATCAAAACGGAGTGA-3']). It is used to compare to the *gcm* morphants. We chose to use a *gcm* morpholino to knockdown *gcm* instead of CRISPR/Cas9 as used previously (*Oulhen and Wessel, 2016*). CRISPR/Cas9 randomly mutates the targeted genomic sequence and in thee wildtype animals we decided it best to not have the variability of different mutations (between experiments, and between each injected embryos) that could have different effects in the gene expression and the phenotypes obtained.

Three Cas9 guide RNAs (gRNAs; 200 ng/ul of each gRNA) previously used in our lab (*Oulhen and Wessel, 2016*) were used to target *pks1* DNA. gRNAs were mixed with 500 ng/ul of Cas9 mRNA, injected into freshly fertilized eggs as described previously in *Oulhen and Wessel, 2016*.

## Phylogenetic analysis

Phylogenetic analyses were conducted using MEGA version 6 (MEGA Software, RRID:SCR_000667) (*Tamura et al., 2011*). Protein sequences were obtained from NCBI and echinobase. Phylogenetic reconstruction was carried out using the maximum likelihood method, and tested also by Neighbor-joining methodology. Both tests resulted in the same outcome. Bootstrap values were determined by 500 replicates. Initial tree(s) for the heuristic search were obtained automatically by applying Neighbor-Join and BioNJ algorithms to a matrix of pairwise distances estimated using a JTT model, and then selecting the topology with superior log likelihood value. The analysis involved 25 amino acid sequences. There were a total of 671 positions in the final dataset. SPU or NCBI accession numbers are as follows: AAA52457 HsFMO; P31513 HsFMO3; Q99518 HsFMO2; CAA77797 HsFMO4; P49326 HsFMO5; SPU_002963 Sp-Fmo2; SPU_007044 Sp-FMO6L; SPU_009114 Sp-Fmo5; SPU_012348 Sp-Fmo5-1; SPU_014947 Sp-Fmo2-2; SPU_017252 Sp-Fmo5-2; SPU_017374 Sp-Fmo3; SPU_017639 Sp-Fmo; SPU_022596 Sp-Fmo5-3; SPU_022597 Sp-Fmo3-1; SPU_022765 Sp-Fmo2-3; SPU_023681 Sp-Fmo2-5; SPU_024227 Sp-Fmo5-6; SPU_024261 Sp-Fmo5-4; SPU_025958 Sp-Fmo2-4; EDL39293.1 Mm-Fmo4; NP_001171509 Mm-Fmo6; BAA03745 Mm-Fmo; AAQ94601 Dr-Fmo1; NP_989910 Gg-Fmo3; AAK94940 Dm-Fmo1; AAL27708 Dm-Fmo2; Q47PU3 Gg-Fmo5; NP_001087387 Xl-Fmo5-1; CAD10798 Comamonas_testosteroni.

## Adult tissues RNA extraction and qPCR

RNA from 100 embryos was isolated with the RNeasy Micro kit (Qiagen, Cat#:74004), while RNA from adult tissues was isolated from Trizol (Thermo Fisher Scientific, Cat#:10296010). cDNA synthesis was performed using Maxima kit (Life Technologies, Cat#:K1641). qPCR was performed using ABI 7900 real time instrument with Maxima SYBR master mix (Life Technologies, Cat#:FERK0222) and normalized to ubiquitin transcripts. Experiments were run in three biological replicates, and every biological replicate was run on the qPCR machine with three technical replicates. In the graphs, the mean of the three technical replicates is shown, and error bars represent biological replicates. List of primers used for qPCR are in *Supplementary file 4*.

## Embryo dissociations

Embryos were collected and washed twice with calcium-free seawater, and then suspended in hyalin-extraction media (HEM) for 15 min (*George and McClay, 2019*). For enrichment of ectodermal cells, embryos were transferred to 0.5M NaCl as soon as the squamous ectodermal epithelium cells became rounded and loosened from each other. The embryos were then gently sheared with a pipette to remove the ectoderm from the basal lamina and the remainder of the embryo was removed from the enriched ectoderm population with a 40 micron Nitex mesh filter. When the entire embryo was to be dissociated, the embryos were subjected to more prolonged HEM treatment, then transferred into 0.5M NaCl, gently sheared with a pipette to complete dissociation, and residual cellular clumps were removed with a 40 micron Nitex mesh. Dissociated cells were counted on a hemocytometer, and diluted with 0.5M NaCl to reach the appropriate concentration for the scRNA-seq protocol. Embryos were collected only by settling at 1xg, and at no time were cells or embryos pelleted in a centrifuge. Throughout the procedure, specimens were kept at 4°C.

## Single cell RNA sequencing

Single cell RNA sequencing: Single cell encapsulation was performed using the Chromium Single Cell Chip B kit on the 10x Genomics Chromium Controller. Single cell cDNA and libraries were

prepared using the Chromium Single Cell 3' Reagent kit v3 Chemistry. Libraries were sequenced by Genewiz on the Illumina Hiseq (RRID:SCR_016387) (2 × 150 bp paired-end runs). The non-injected 48hpf and 72hpf samples were sequenced in separate lanes (350M reads per lane). The morpholino injected samples were sequenced together in the same lane (350M reads total). Single cell unique molecular identifier (UMI) counting was performed using Cell Ranger Single Cell Software Suite 3.0.2 from 10X Genomics (Cell Ranger, RRID:SCR_017344). The custom transcriptome reference was generated from assembly Spur_4.2 using Cell Ranger. Cell Ranger matrices were further analyzed using the R package Seurat v 3.1.4 (SEURAT, RRID:SCR_007322) (*Stuart et al., 1821*). Cells with at least 400 and at most 2500 different represented genes were included in downstream analysis. The top 2000 highly variable gene representatives (features) across the datasets were used to integrate datasets. TSNE projection and clustering analysis for all datasets were conducted using 15 dimensions and a resolution of 0.5. Cluster markers were found using Find Conserved Markers and Find Markers functions. The integration of gcm-enriched clusters from multiple time points was performed using Seurat following the standard integration pipeline (dims:10 res:0.1). The time course dataset included samples collected from eight-cell stage, 64 cell stage, morula, early blastula, hatched blastula, mesenchyme blastula, early gastrula and late gastrula (4-8hpf) stage (*Foster et al., 2020*). The sequencing data generated here have been made publicly available at Gene Expression Omnibus [https://www.ncbi.nlm.nih.gov/geo/] (GSE155427).

## Acknowledgements

We are grateful for, and acknowledge the support for this work from the National Institutes of Health grants 9RO1GM125071 and 1R01GM132222 to GMW and 1P20GM119943 to NO. The Sp1 antibody developed by Robert Burke was obtained from the Developmental Studies Hybridoma Bank, created by the NICHD of the NIH and maintained at The University of Iowa, Department of Biology, Iowa City, IA 52242; https://dshb.biology.uiowa.edu/. We are also grateful for use of the resources and services at the omputational Biology Core and Center for Computation and Visualization, Brown University.

## Additional information

### Funding

| Funder | Grant reference number | Author |
|---|---|---|
| National Institutes of Health | 9RO1GM125071 | Gary Wessel |
| National Institutes of Health | 1R01GM132222 | Gary Wessel |
| National Institutes of Health | 1P20GM119943 | Nathalie Oulhen |

The funders had no role in study design, data collection and interpretation, or the decision to submit the work for publication.

### Author contributions

Margherita Perillo, Conceptualization, Data curation, Formal analysis, Validation, Investigation, Visualization, Methodology, Writing - original draft, Writing - review and editing; Nathalie Oulhen, Conceptualization, Software, Formal analysis, Validation, Investigation, Methodology, Writing - review and editing; Stephany Foster, Data curation, Software, Formal analysis, Investigation, Methodology, Writing - review and editing; Maxwell Spurrell, Formal analysis, Investigation, Methodology, Writing - review and editing; Cristina Calestani, Conceptualization, Formal analysis, Writing - review and editing; Gary Wessel, Conceptualization, Data curation, Formal analysis, Supervision, Funding acquisition, Investigation, Visualization, Methodology, Writing - original draft, Project administration, Writing - review and editing

### Author ORCIDs

Margherita Perillo https://orcid.org/0000-0003-0845-507X
Stephany Foster https://orcid.org/0000-0002-9404-1692

Maxwell Spurrell  https://orcid.org/0000-0001-7314-7482
Cristina Calestani  https://orcid.org/0000-0001-8122-9349
Gary Wessel  https://orcid.org/0000-0002-1210-9279

**Decision letter and Author response**
Decision letter https://doi.org/10.7554/eLife.60388.sa1
Author response https://doi.org/10.7554/eLife.60388.sa2

## Additional files

### Supplementary files

• Supplementary file 1. Example of cluster 12 comparisons to clusters 13 at 48 hr (12 vs 13 at 48 hr) and at 72 hr (12 vs 13 at 72 hr) shows that three pigment cell genes (*pks1, sult1c2, fmo5*) are less expressed in cluster 12 than 13.

• Supplementary file 2. Comparisons of clusters 2 versus 13. Negative values indicate genes enriched in cluster 13 compared to cluster 2.

• Supplementary file 3. List of primers used for in situ hybridization probes.

• Supplementary file 4. List of primers used for qPCRs.

• Transparent reporting form

### Data availability

The sequencing data have been made publicly available at Gene Expression Omnibus under GSE155427.

The following dataset was generated:

| Author(s) | Year | Dataset title | Dataset URL | Database and Identifier |
|---|---|---|---|---|
| Foster S, Oulhen N, Wessel GM | 2020 | Sp single cell RNA-seq | http://www.ncbi.nlm.nih.gov/geo/query/acc.cgi?acc=GSE155427 | NCBI Gene Expression Omnibus, GSE155427 |

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
