## [Decision Letter]

**Acceptance summary:**

This manuscript uses single-cell sequencing to characterize the development of pigment cells in the purple sea urchin. In addition to producing various chemically interesting pigments, these cells play a role in the animal's innate immune system (e.g., one of the pigments has been shown to have anti-microbial activity). This work presents a new view of the process of pigment cell development in the purple sea urchin, clarifies the relationships between larval pigment cells and different adult pigmented cell types and some immune cells, and will serve as a useful resource for the field.

**Decision letter after peer review:**

Thank you for submitting your article "Regulation of dynamic pigment cell states at single-cell resolution" for consideration by *eLife*. Your article has been reviewed by three peer reviewers, one of whom is a member of our Board of Reviewing Editors, and the evaluation has been overseen by Marianne Bronner as the Senior Editor. The following individual involved in review of your submission has agreed to reveal their identity: Jonathan Patrick Rast (Reviewer #2).

The reviewers have discussed the reviews with one another and the Reviewing Editor has drafted this decision to help you prepare a revised submission.

Summary:

This manuscript by Perillo et al. uses single-cell sequencing to characterize the development of pigment cells in the purple sea urchin, *Strongylocentrotus purpuratus*. In addition to producing various chemically interesting pigments, these cells play a role in the animal's innate immune system (e.g., one of the pigments has been shown to have anti-microbial activity). Thus, understanding how these cells are generated and characterizing the biochemical pathways used to synthesize their diverse pigments should be of broad interest.

Single-cell analysis at distinct developmental time points (gastrulae and larvae) identified two pigment cell clusters in larval ectoderm and one cluster of mesodermal pigment cell precursors in gastrulae. One of the larval pigment cell clusters expresses cell cycle-associated genes and the authors show that cells expressing a pigment cell marker can incorporate EdU in gastrulae and larvae. New pigment cell markers identified via single-cell sequencing were validated by double fluorescent in situ hybridization. Because a flavin-dependent monooxygenase (*fmo*) is involved in pigment synthesis, the authors examined *fmo* genes present in the *S. purpuratus* genome, and found one group (of the four families they describe) with enriched expression in embryonic pigment cells and adult coelomocytes (immune cells). Finally, based on previous work showing that the transcription factor glial cells missing (*gcm*) was required for pigmentation, the authors perform single-cell analysis of *gcm* knockdown embryos and conclude that this gene is required for pigment cell specification. Overall, this work presents a new view of the process of pigment cell development in the purple sea urchin as well as the relationship of larval pigment cells to different adult pigment cell types and to red spherule immune cells. Pending the clarification of the issues outlined below and substantial editing of the text, it will serve as a useful resource for those interested in this system.

Essential revisions:

1) The authors seem to be overstating the case that they have identified a population of pigment stem cells. It has been evident for some time that pigment cell numbers increase after they delaminate from the archenteron and before they differentiate (Gibson and Burke, 1987) consistent with a round of cell division after delamination. What evidence is there from the single cell sequence data that these cells have stem cell qualities and are not just progenitors at the last division before differentiation. Their data show that pigment cells can divide, but their subsequent fates after taking up EdU have not been followed. In the absence of additional evidence, the authors should soften their language about stem cells and stem cell niches.

2) Do the authors know whether the dividing Cluster 13 cells are granular at this point in development? A description of the morphology/behavior of the cluster 12 and cluster 13 cells would be interesting if the authors know anything about this. Cluster 12 is of interest since it is most unlike cluster 2 and 13. As GCM disappears at 72 hrs in these cells, the cluster remains in the tSNE plot. Do the authors know where these cells are at this time? Are these some type of blastocoelar cell?

3) In their characterization of *fmo* gene families in *S. purpuratus*, the authors conclude that two of the protein families are not found in other organisms. Because the only animals represented are humans and *S. purpuratus* (Figure 4A), it would be reasonable for the authors to broaden the sampling of organisms on their phylogenetic tree to include a more diverse set of animals.

4) Because the authors have previously published the lack of pigment cells following CRISPR/Cas9 targeting of *gcm* (Oulhen and Wessel, 2016), they should clarify why they instead used morpholino knockdowns for the scSeq experiments reported here.

5) In Figure 6F it is not obvious that pigment cell numbers are unaffected by PKS Cas9. Based on the image shown, a naïve reader could conclude that there are fewer pigment cells and that they are larger. As they have done for *gcm* MO embryos, this should be quantified. It would also help the reader if the rationale for this experiment were made clearer at the outset. In its current form the results are described in the middle of a paragraph about the phenotype of *gcm* knockdown embryos without any introduction or explanation.

6) What is covered in Foster et al., 2020, in review? How is it non-overlapping with this paper? What is the significance of the early non-pigment cell GCM expression in Figure 2? It is difficult to see the data in detail here and this may be important since the finding seems to contradict earlier in situ data, FP transgene analysis and transcript quantification data. It may be that single-cell sequencing can discern sporadic transcription that is seen as background in previous studies but more information on estimated expression levels would be useful here. Is this seen for other well characterized transcription factors in the developmental scRNA seq data?

7) The rationale and description of integrating this *gcm* expression analysis with the gastrula and larva scSeq (starting at the second paragraph of the subsection “Gcm is Enriched in Three Clusters at Gastrula and Early Larva Stages”) was unclear. For example, the reader is told that following integration of the various time points, "one cluster is marked by the expression of *gcm*…", but this starting point is never shown. It may be useful for the reader if somewhere in the figure legends, the 7 detected cell states could be defined, since they are all numbered in the figures. The authors do a very nice job of presenting relevant numbers in other figures (of cells sequenced, average number of reads, average number of UMI/cell), but this section is very unclear on where the data are coming from: how many total cells? what percentage express *gcm*? etc…

8) To help the authors clarify their presentation, the extensive editorial comments of reviewer #3 (included below) should be addressed in the revised manuscript.

---

## [Author Response]

Essential revisions:1) The authors seem to be overstating the case that they have identified a population of pigment stem cells. It has been evident for some time that pigment cell numbers increase after they delaminate from the archenteron and before they differentiate (Gibson and Burke 1987) consistent with a round of cell division after delamination. What evidence is there from the single cell sequence data that these cells have stem cell qualities and are not just progenitors at the last division before differentiation. Their data show that pigment cells can divide, but their subsequent fates after taking up EdU have not been followed. In the absence of additional evidence, the authors should soften their language about stem cells and stem cell niches.

We have tempered the language of pigment stem cells in the Abstract, Introduction, and Results, but we have retained this consideration in the Discussion section to ensure a broader dialog within the community

2) Do the authors know whether the dividing Cluster 13 cells are granular at this point in development? A description of the morphology/behavior of the cluster 12 and cluster 13 cells would be interesting if the authors know anything about this. Cluster 12 is of interest since it is most unlike cluster 2 and 13. As GCM disappears at 72 hrs in these cells, the cluster remains in the tSNE plot. Do the authors know where these cells are at this time? Are these some type of blastocoelar cell?

The cluster 13 cells are not blastocoelar cells because we did not find any blastocoelar markers (e.g. ese, gataC). Cluster 12 has mesodermal markers and might represent undifferentiated mesoderm. From Sp1 staining they look migratory, extending filopodia as the non-dividing cells, but we cannot yet identify pigment levels in these cells.

3) In their characterization of fmo gene families in S. purpuratus, the authors conclude that two of the protein families are not found in other organisms. Because the only animals represented are humans and S. purpuratus (Figure 4A), it would be reasonable for the authors to broaden the sampling of organisms on their phylogenetic tree to include a more diverse set of animals.

Good point – We made a new phylogenetic tree of *fmos* including sequences from mouse, frog, fish, flies, chicken. In this new tree we found that Fmo5-6 belongs to clade 3 (which previously did not group with any other sequence), but all other associations were consistent. We now document that two different methods of tree construction (maximum likelihood and maximum parsimony) resulted in the same outcome, providing additional confidence in our interpretations.

4) Because the authors have previously published the lack of pigment cells following CRISPR/Cas9 targeting of gcm (Oulhen and Wessel, 2016), they should clarify why they instead used morpholino knockdowns for the scSeq experiments reported here.

CRISPR/Cas9 randomly mutates the targeted genomic sequences. These are, after all, wildtype animals and not strains of homogenous animals that have a single type of mutation. We wanted to have data that could be clearly interpreted and reproducible and the morpholino will always bind the same sequence of the mRNA to stop its translation. Every injection of CRISPR/Cas9, however, will yield different mutations (between experiments, and between each injected embryo) that could have different effects in the gene expression and the phenotypes obtained. Moreover, with CRISPR/Cas9, the resulting truncated proteins are still being translated and could still be active depending on the mutations. Such variations may complicate the scRNA-sq results so we decided to use the morpholino as a more reliable method to downregulate *gcm*. A paraphrased version of this explanation is now in the Materials and methods section.

5) In Figure 6F it is not obvious that pigment cell numbers are unaffected by PKS Cas9. Based on the image shown, a naïve reader could conclude that there are fewer pigment cells and that they are larger. As they have done for gcm MO embryos, this should be quantified. It would also help the reader if the rationale for this experiment were made clearer at the outset. In its current form the results are described in the middle of a paragraph about the phenotype of gcm knockdown embryos without any introduction or explanation.

The Pks1 – KO here is used to test homeostasis in the pigment cells and whether PKS may have any function, directly or indirectly, in regulating gene expression. This is topical in the Discussion. We have now also mentioned this briefly in the Results section as justification for doing the experiment, but retained its position in the paragraph. It is an experimental result that contrasts with the GCM experiment, so we believe the order in the paragraph is appropriate. We have now also quantitated these results as requested and these data are now shown in Figure 6F, G and H.

6) What is covered in Foster et al., 2020, in review? How is it non-overlapping with this paper? What is the significance of the early non-pigment cell GCM expression in Figure 2? It is difficult to see the data in detail here and this may be important since the finding seems to contradict earlier in situ data, FP transgene analysis and transcript quantification data. It may be that single-cell sequencing can discern sporadic transcription that is seen as background in previous studies but more information on estimated expression levels would be useful here. Is this seen for other well characterized transcription factors in the developmental scRNA seq data?

Foster et al., 2020, presents a single cell RNA-seq resource dataset to the community of 8-timepoints of *S. purpuratus* development from 8 – cell stage to 48 hours. We use that dataset here to extract *gcm* mRNA expression, which was not performed in Foster et al., 2020. As seen for several genes, the mRNA profile is more broad and earlier than seen by in situ RNA or qPCR experiments. Foster et al., 2020, discusses this phenomenon extensively, and we refer to that discussion but only briefly highlight the finding here. The Foster et al., 2020 manuscript was recently accepted for publication (Development) so readers of this pigment paper will also have access to that single cell RNA-seq data.

7) The rationale and description of integrating this gcm expression analysis with the gastrula and larva scSeq (starting at the second paragraph of the subsection “Gcm is Enriched in Three Clusters at Gastrula and Early Larva Stages”) was unclear. For example, the reader is told that following integration of the various time points, "one cluster is marked by the expression of gcm…", but this starting point is never shown. It may be useful for the reader if somewhere in the figure legends, the 7 detected cell states could be defined, since they are all numbered in the figures. The authors do a very nice job of presenting relevant numbers in other figures (of cells sequenced, average number of reads, average number of UMI/cell), but this section is very unclear on where the data are coming from: how many total cells? what percentage express gcm? etc…

Gcm is first detected at 64-cell stage, when there are 9 *gcm*+ cells, 1% of total cells. At morula stage 125 cells express *gcm*, 3.48%. Early blastula:13.9% Hatched: 9.5% Mesenchyme blastula: 7.6% Early gastrula: 6.5% and late gastrula 4.7%. This result is now presented in the figure legend of Figure 2.

8) To help the authors clarify their presentation, the extensive editorial comments of reviewer #3 (included below) should be addressed in the revised manuscript.

Thank you for this effort. We have now done so and have addressed each issue below.